# STaSy: Score-based Tabular data Synthesis

**Jayoung Kim, Chaejeong Lee, and Noseong Park**
Department of Artificial Intelligence
Yonsei University
Seoul, South Korea
`{jayoung.kim, chaejeong_lee, noseong}@yonsei.ac.kr`

## ABSTRACT

Tabular data synthesis is a long-standing research topic in machine learning. Many different methods have been proposed over the past decades, ranging from statistical methods to deep generative methods. However, it has not always been successful due to the complicated nature of real-world tabular data. In this paper, we present a new model named **S**core-based **Ta**bular data **Sy**nthesis (`STaSy`) and its training strategy based on the paradigm of score-based generative modeling. Despite the fact that score-based generative models have resolved many issues in generative models, there still exists room for improvement in tabular data synthesis. Our proposed training strategy includes a self-paced learning technique and a fine-tuning strategy, which further increases the sampling quality and diversity by stabilizing the denoising score matching training. Furthermore, we also conduct rigorous experimental studies in terms of the generative task trilemma: sampling quality, diversity, and time. In our experiments with 15 benchmark tabular datasets and 7 baselines, our method outperforms existing methods in terms of task-dependant evaluations and diversity.

## 1 INTRODUCTION

Tabular data synthesis is of non-trivial importance in real-world applications for various reasons: protecting the privacy of original tabular data by releasing fake tabular data (Park et al., 2018; Lee et al., 2021), augmenting the original tabular data with fake data for better training machine learning models (Chawla et al., 2002; Han et al., 2005; He et al., 2008; Kim et al., 2022), and so on. However, it is well-known that tabular data frequently has such peculiar characteristics that deep generative models are not able to synthesize all possible details of the original tabular data (Park et al., 2018; Xu et al., 2019) — given a set of columns in tabular data, columns typically follow unpredictable (multi-modal) distributions and therefore, it is hard to model their joint probability.

Table 1: Summary of experimental results. We report the average sampling quality, diversity, and time.

| Methods | Quality ↑ (F1 & $R^2$) | Diversity ↑ (coverage) | Runtime ↓ (second) |
|---|---|---|---|
| MedGAN | -0.717 | 0.037 | 0.246 |
| VEEGAN | -0.368 | 0.038 | 0.109 |
| CTGAN | 0.560 | 0.352 | 0.704 |
| TVAE | 0.474 | 0.494 | 0.100 |
| TableGAN | 0.386 | 0.434 | **0.046** |
| OCT-GAN | 0.552 | 0.381 | 26.926 |
| RNODE | 0.366 | 0.328 | 13.392 |
| Naïve-STaSy | 0.708 | 0.637 | 8.855 |
| STaSy | **0.727** | **0.658** | 10.663 |

A couple of recent methods, however, showed remarkable successes (with some failure cases) in synthesizing fake tabular data, such as CTGAN (Xu et al., 2019), TVAE (Xu et al., 2019), IT-GAN (Lee et al., 2021), and OCT-GAN (Kim et al., 2021). In addition, a recent generative model paradigm, called score-based generative modeling (SGMs), successfully resolves the two problems of *the generative learning trilemma* (Xiao et al., 2021), i.e., score-based generative models provide high sampling quality and diversity, although their training/sampling time is relatively longer than other deep generative models. In this paper, we adopt a score-based generative modeling paradigm and design a **S**core-based **Ta**bular data **Sy**nthesis (`STaSy`) method.

Our model designs significantly outperform all existing baselines in terms of the sampling quality and diversity (cf. `Naïve-STaSy` and `STaSy` in Table 1) — `Naïve-STaSy` is a naive conversion

of SGMs toward tabular data, and `STaSy` additionally uses our proposed self-paced learning and fine-tuning methods. Figure 1 shows the uneven and long-tailed loss distribution of `Naïve-STaSy` at the end of its training process. The figure implies the training of `Naïve-STaSy` by the denoising score matching failed to learn the score values of some records. This may let the model be (partially) underfitted to training data. In contrast, `STaSy` with our two proposed training methods yields many loss values around the left corner (i.e., close to 0).

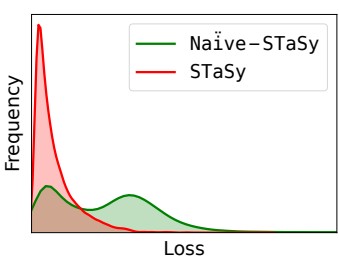

Figure 1: Distributions of denoising score matching loss in `Shoppers`

In order to alleviate the training difficulty of `Naïve-STaSy`, we design i) a self-paced learning method, and ii) a fine-tuning approach. Our proposed self-paced learning technique trains our model from easy to hard records based on their loss values by modifying the objective function. The technique makes the model learn records selectively and eventually. During this process, the model can be better trained. In addition, our proposed fine-tuning method, which modestly adjusts the model parameters, can further improve the sampling quality and diversity.

In Table 1, we summarize our experimental results, where we compare our `STaSy` with other existing tabular data synthesis methods in terms of the sampling quality, diversity, and time. As shown, our basic model even without our proposed self-paced learning and fine-tuning, denoted `Naïve-STaSy`, significantly outperforms all baselines except for runtime.

In summary, our contributions are as follows: i) We design a score-based generative model for tabular data synthesis. ii) We alleviate the training difficulty of the denoising score matching loss by designing a self-paced learning strategy and further enhance the sampling quality and diversity using a proposed fine-tuning method. `STaSy`, thus, clearly balances among *the generative learning trilemma*: sampling quality, diversity, and time. iii) Our proposed method outperforms other deep learning methods in all cases by large margins, which we consider a significant advance in the field of tabular data synthesis. iv) We evaluate various methods in terms of *the generative learning trilemma* in a rigorous manner.

## 2 RELATED WORK

### 2.1 SCORE-BASED GENERATIVE MODELS

Score-based generative models (SGMs) use a diffusion process defined by the following Itô stochastic differential equation (SDE):

$$d\mathbf{x} = \mathbf{f}(\mathbf{x}, t)dt + g(t)d\mathbf{w}, \tag{1}$$

where $\mathbf{f}(\mathbf{x}, t) = f(t)\mathbf{x}$, $f$ and $g$ are drift and diffusion coefficients of $\mathbf{x}(t)$, and $\mathbf{w}$ is the standard Wiener process. Depending on the types of $f$ and $g$, SGMs can be divided into variance exploding (VE), variance preserving (VP), and sub-variance preserving (sub-VP) models (Song et al., 2021). The definitions of $f$ and $g$ are in Appendix A. The reverse of the diffusion process is a denoising process as follows:

$$d\mathbf{x} = \big(\mathbf{f}(\mathbf{x}, t) - g^2(t)\nabla_{\mathbf{x}} \log p_t(\mathbf{x})\big)dt + g(t)d\mathbf{w}, \tag{2}$$

where this reverse SDE is a process of generating samples. The score function $\nabla_{\mathbf{x}} \log p_t(\mathbf{x})$ is approximated by a time-dependent score-based model $S_{\boldsymbol{\theta}}(\mathbf{x}, t)$, called *score network*.

In general, following the diffusion process in Equation 1, we can derive $\mathbf{x}(t)$ at time $t \in [0, T]$, where $\mathbf{x}(0)$ and $\mathbf{x}(T)$ means a real and noisy sample, respectively. The transition probability $p(\mathbf{x}(t)|\mathbf{x}(0))$ at time $t$ is easily approximated by this process, and it always follows a Gaussian distribution. It allows us to collect the gradient of the log transition probability, $\nabla_{\mathbf{x}(t)} \log p(\mathbf{x}(t)|\mathbf{x}(0))$, during the diffusion process. Therefore, we can train a score network $S_{\boldsymbol{\theta}}(\mathbf{x}, t)$ as follows:

$$\arg \min_{\boldsymbol{\theta}} \mathbb{E}_t \mathbb{E}_{\mathbf{x}(t)} \mathbb{E}_{\mathbf{x}(0)} \Big[\lambda(t)\|S_{\boldsymbol{\theta}}(\mathbf{x}(t), t) - \nabla_{\mathbf{x}(t)} \log p(\mathbf{x}(t)|\mathbf{x}(0))\|_2^2\Big], \tag{3}$$

where $\lambda(t)$ is to control the trade-off between the sampling quality and likelihood. This is called *denoising score matching*, and $\boldsymbol{\theta}^*$ solving Equation 3 can accurately solve the reverse SDE in Equation 2 (Vincent, 2011).

After the training process, we can synthesize fake data records with i) the *predictor-corrector* framework or ii) the *probability flow* method, a deterministic method based on the ordinary differential equation (ODE) whose marginal distribution is equal to that of Equation 1 (Song et al., 2021). In particular, the latter enables fast sampling and exact log-probability computation.

## 2.2 TABULAR DATA SYNTHESIS

Many distinct methods exist for tabular data synthesis, which creates realistic synthetic tables depending on the data types. For example, a recursive table modeling utilizing a Gaussian copula is used to synthesize continuous variables (Patki et al., 2016). Discrete variables can be generated by Bayesian networks (Zhang et al., 2017; Aviñó et al., 2018) and decision trees (Reiter, 2005). Several data synthesis methods based on GANs have been presented to generate tabular data in recent years. RGAN (Esteban et al., 2017) creates continuous time-series healthcare records, whereas `MedGAN` (Choi et al., 2017) and corrGAN (Patel et al., 2018) generate discrete records. EhrGAN (Che et al., 2017) utilizes semi-supervised learning to generate plausible labeled records to supplement limited training data. PATE-GAN (Jordon et al., 2019) generates synthetic data without jeopardizing the privacy of real data. `TableGAN` (Park et al., 2018) employs convolutional neural networks to enhance tabular data synthesis and maximize label column prediction accuracy. `CTGAN` and `TVAE` (Xu et al., 2019) adopt column-type-specific preprocessing steps to deal with multi-modality in the original dataset distribution. `OCT-GAN` (Kim et al., 2021) is a generative model design based on neural ODEs. `SOS` (Kim et al., 2022) proposed a style-transfer-based oversampling method for imbalanced tabular data using SGMs, whose main strategy is converting a major sample to a minor sample. Since its task is not compatible to our task to generate from the scratch, direct comparisons are not possible. However, we convert our method to an oversampling method following their design guidance and compare with `SOS` in Appendix B.

## 2.3 SELF-PACED LEARNING

Self-paced learning (SPL) is a training strategy related to curriculum learning to select training records in a meaningful order, inspired by the learning process of humans (Kumar et al., 2010b; Jiang et al., 2014). It refers to training a model only with a subset of data that has low training losses and gradually expanding to the entire training data. We denote the training set as $\mathcal{D} = \{\mathbf{x}_i\}_{i=1}^N$, where $\mathbf{x}_i$ is the $i$-th record. The model $M$ with parameters $\boldsymbol{\theta}$ has a loss $l_i = L(M(\mathbf{x}_i, \boldsymbol{\theta}))$, where $L$ is the loss function. A vector $\mathbf{v} = [v_i]_{i=1}^N, v_i \in \{0, 1\}$ indicates whether $\mathbf{x}_i$ is easy or not for all $i$. SPL aims to learn the model parameter $\boldsymbol{\theta}$ and the selection importance $\mathbf{v}$ by minimizing:

$$\min_{\boldsymbol{\theta}, \mathbf{v}} \mathbb{E}(\boldsymbol{\theta}, \mathbf{v}) = \sum_{i=1}^N v_i L(M(\mathbf{x}_i, \boldsymbol{\theta})) - \frac{1}{K} \sum_{i=1}^N v_i, \tag{4}$$

where $K$ is a parameter to control the learning pace. In general, the second term in Equation 4, called a self-paced regularizer, can be customized for a downstream task.

The alternative convex search (ACS) (Bazaraa et al., 1993) is typically used to solve Equation 4 (Kumar et al., 2010a; Tang et al., 2012). By alternately optimizing variables while fixing others, we can optimize Equation 4, i.e., update $\mathbf{v}$ after fixing $\boldsymbol{\theta}$, and vice versa. With fixed $\boldsymbol{\theta}$, the global optimum $\mathbf{v}^* = [v_i^*]_{i=1}^N$ is defined as follows:

$$v_i^* = \begin{cases} 1, & l_i < \frac{1}{K}, \\ 0, & l_i \geq \frac{1}{K}, \end{cases} \tag{5}$$

When updating $\mathbf{v}$ with fixed $\boldsymbol{\theta}$, a record $\mathbf{x}_i$ with $l_i < \frac{1}{K}$ is regarded as an easy record and will be chosen for training. Only easy records are used to train the model. Otherwise, $\mathbf{x}_i$ is regarded as a hard record and will be unselected. To involve more records in the training process, $K$ is gradually decreased.

## 3 PROPOSED METHOD

`STaSy` is an SGM-based method for tabular data synthesis. `STaSy` uses SPL to ensure its training stability. The suggested fine-tuning method takes advantage of a favorable property of SGMs, which is that we can measure the log-probabilities of records.

### 3.1 SCORE NETWORK ARCHITECTURE & MISCELLANEOUS DESIGNS

It is known that each column in tabular data typically has complicated distributions, whereas pixel values in image datasets typically follow Gaussian distributions (Xu et al., 2019). Moreover, tabular synthesis models should learn the joint probability of multiple columns to generate a record, which is one main reason why tabular data synthesis is difficult. However, one good design point is that the dimensionality of tabular data is typically far less than that of image data, e.g., 784 pixels even in MNIST, one of the simplest image datasets, vs. 30 columns in `Credit`.

We found through our preliminary experiments that the SDE in Equations 1 and 2 can well model the joint probability *iff* its score network, which approximates $\nabla_{\mathbf{x}} \log p_t(\mathbf{x})$, is well trained. We carefully design our score network for tabular data synthesis considering these points. Our proposed score network architecture is in Appendix C. The network consists of residual blocks of FC layers.

Since SGMs were theoretically designed from the idea of perturbing data with an infinite number of noise scales, SGMs typically require large-scale computation, e.g., $T = 1,000$ for images in (Song et al., 2021), as an approximation to the infinite number. With a large number of steps, the denoising process requires a long time to complete, which is one part of *the generative learning trilemma*. However, we found that $T = 50$ steps in Equation 1 are enough to train a network to approximate the gradient of the log-likelihood, which means that our `STaSy` naturally has less sampling time than SGMs for images with $T = 1,000$ steps.

**Pre/post-processing of tabular data** To handle mixed types of data, which is a challenge in tabular data generation, we pre/post-process columns. We use the min-max scaler to pre-process numerical columns, and its reverse scaler is used for post-processing after generation. We also apply one-hot encoding to pre-process categorical columns, and use the softmax function, followed by the rounding function, when generating.

**How to generate** After sampling a noisy vector $\mathbf{z} \sim \mathcal{N}(\boldsymbol{\mu}, \sigma^2 \mathbf{I})$, the reverse SDE can convert $\mathbf{z}$ into a fake record. The prior distribution $\mathbf{z} \sim \mathcal{N}(\boldsymbol{\mu}, \sigma^2 \mathbf{I})$ varies depending on the type of SDEs: $\mathcal{N}(\mathbf{0}, \sigma_{max}^2 \mathbf{I})$ for VE, and $\mathcal{N}(\mathbf{0}, \mathbf{I})$ for VP and sub-VP. $\sigma_{max}$ is a hyperparameter. In particular, we adopt the *probability flow* method to solve the reverse SDE, which will be shortly described in Equation 10.

### 3.2 SELF-PACED LEARNING APPROACH

In order to alleviate the training difficulty, we apply a curriculum learning technique for `STaSy`, more specifically, self-paced learning. Instead of letting $v_i \in \{0, 1\}$, we use a "soft" record sampling method, i.e., $v_i \in [0, 1]$. If $l_i$, which is the denoising score matching objective on $i$-th record, is less than a threshold, we set $v_i$ to 1 to ensure that the record is fully involved in training. At the end of the training, $v_i$ must be set to 1 for all $i$ to train the model with the entire data. The denoising score matching loss for $i$-th training record $\mathbf{x}_i$ is defined as follows:

$$l_i = \mathbb{E}_t \mathbb{E}_{\mathbf{x}_i(t)} \left[ \lambda(t) \| S_{\boldsymbol{\theta}}(\mathbf{x}_i(t), t) - \nabla_{\mathbf{x}_i(t)} \log p(\mathbf{x}_i(t) | \mathbf{x}_i(0)) \|_2^2 \right]. \tag{6}$$

Then, we have the following `STaSy` objective:

$$\min_{\boldsymbol{\theta}, \mathbf{v}} \sum_{i=1}^{N} v_i l_i + r(\mathbf{v}; \alpha, \beta), \tag{7}$$

where $0 \le v_i \le 1$ for all $i$, $r(\cdot)$ is a self-paced regularizer. $\alpha \in [0, 1]$ and $\beta \in [0, 1]$ are variables to define thresholds, which are monotonically increasing as training goes on (see Appendix D for their exact controlling mechanism).

**Definition 1.** *Let $Q(p)$ be a quantile function defined as $\inf\{l \in \mathbb{R} : p \le F(l)\}$, where $F$ is a cumulative distribution function of the denoising score matching objective. That is, $Q(p)$ is the minimum value for which the CDF is greater than or equal to the given probability $p$.*

---

**Algorithm 1:** How to train `STaSy`

---

1 Initialize $\boldsymbol{\theta}$
  /* Train SGM based on our SPL training strategy                */
2 **for** *each mini-batch of records* **do**
3      Update $\mathbf{v}$ with Equation 9
4      Update $\boldsymbol{\theta}$ after fixing $\mathbf{v}$ with Equation 7
5      Update $\alpha$ and $\beta$ with the control method in Appendix D
  /* Fine-tune the trained model using log-probability          */
6 $\tau_i \leftarrow \log p(\mathbf{x}_i)$
7 $\mathcal{F} \leftarrow \{\mathbf{x}_i | \log p(\mathbf{x}_i)$, where $\mathbf{x}_i \in \mathcal{D}$, is smaller than the average (or median) log-probability.$\}$
8 **for** *each fine-tune epoch* **do**
9      **for** *each* $\mathbf{x}_i \in \mathcal{F}$ **do**
10          Update $\boldsymbol{\theta}$ with Equation 6
11      $\mathcal{F} \leftarrow \{\mathbf{x}_i | \log p(\mathbf{x}_i) < \tau_i\}$
12 **return** $\boldsymbol{\theta}$

---

**Theorem 1.** *Let the self-paced regularizer* $r(\mathbf{v}; \alpha, \beta)$ *be defined as follows:*

$$r(\mathbf{v}; \alpha, \beta) = -\frac{Q(\alpha) - Q(\beta)}{2} \sum_{i=1}^{N} v_i^2 - Q(\beta) \sum_{i=1}^{N} v_i, \tag{8}$$

*where the closed-form optimal solution for* $\mathbf{v}^* = [v_1^*, v_2^*, \ldots, v_N^*]$, *given fixed* $\boldsymbol{\theta}$, *is defined as follows — its proof is in Appendix E:*

$$v_i^* = \begin{cases} 1, & \textit{if } l_i \leq Q(\alpha), \\ 0, & \textit{if } l_i \geq Q(\beta), \\ \dfrac{l_i - Q(\beta)}{Q(\alpha) - Q(\beta)}, & \textit{otherwise.} \end{cases} \tag{9}$$

Specifically, records with $l_i \leq Q(\alpha)$ are considered easy records and will be selected for training, whereas records with $l_i \geq Q(\beta)$ are considered complicated (or potentially noisy) and will not be selected. If not both cases, records will be partially selected during training, i.e., $v_i \in [0, 1]$. $\alpha$ and $\beta$ are gradually increased to 1 from the initial values $\alpha_0$ and $\beta_0$, proportionally to training progress to ensure that all data records are involved in training. As $\alpha$ and $\beta$ increase, the difficult records are gradually involved in training, and the model also becomes more robust to those difficult cases. We set $\alpha_0$ and $\beta_0$ in such a way that more than 80% of the training records are included in the learning process from the beginning.

### 3.3 FINE-TUNING APPROACH

For solving the reverse SDE process, score-based generative models rely on various numerical approaches. One of the techniques is the *probability flow* method in Equation 10 (Song et al., 2021), which uses a deterministic process whose marginal probability is the same as the SDE. With the approximated score function $S_{\boldsymbol{\theta}}(\cdot)$, the *probability flow* method uses the following neural ordinary differential equation (NODE) based model (Chen et al., 2018):

$$d\mathbf{x} = \left(\mathbf{f}(\mathbf{x}, t) - \frac{1}{2} g(t)^2 \nabla_{\mathbf{x}} \log p_t(\mathbf{x})\right) dt. \tag{10}$$

In our experiments, the *probability flow* shows better quality than other methods to solve the original reverse SDE, and our default solver is the *probability flow* (see Section 4.3). In addition, NODEs facilitate computing the log-probability defined in Equation 10 through the instantaneous change of variables theorem. Consequently, we can calculate the exact log-probability efficiently with the unbiased Hutchinson's estimator (Hutchinson, 1989; Grathwohl et al., 2018). Thus, we propose to fine-tune based on the exact log-probability.

After learning the model parameter $\boldsymbol{\theta}$ as described in Section 3.2, we set the sample-wise threshold $\tau_i$ to $\log p(x_i)$ (cf. Line 6 of Algorithm 1). We then prepare the fine-tuning candidate set $\mathcal{F}$ (cf. Line 7

of Algorithm 1). After fine-tuning for the samples in $\mathcal{F}$, we update the candidate set (cf. Line 11 of Algorithm 1). Our goal is for achieving a better log-probability than the initial one $\tau_i$ before the fine-tuning process.

## 3.4 TRAINING ALGORITHM

Algorithm 1 shows the overall training process for our `STaSy`. Firstly, we initialize the parameters of the score network $\boldsymbol{\theta}$. Utilizing ACS, we then train `STaSy` with the SPL training strategy. At this step, we iteratively optimize $\boldsymbol{\theta}$ and $\mathbf{v}$. We can obtain an optimal $\boldsymbol{\theta}$ by optimizing Equation 7 with fixed $\mathbf{v}$, and the global optimum $\mathbf{v}$ is calculated by Equation 9. We also update $\alpha$ and $\beta$ in proportion to training progress. After finishing the main SPL training step, we can generate fake records from the model. To further improve the trained score network, we retrain our model for every record $\mathbf{x}_i$ whose log-probabilities are less than its threshold $\tau_i$. At Line 10 of Algorithm 1, one can use the log-probability instead of Equation 6 as a fine-tuning objective. However, we found that Equation 6 is more effective (see Appendix F).

## 4 EXPERIMENTS

We analyze methods in terms of *the generative learning trilemma*. We list only aggregated results over datasets in the main paper — details are in Appendix G. We repeat experiments 5 times.

### 4.1 EXPERIMENTAL ENVIRONMENTS

A brief description of our experimental environments is as follows: i) We use 15 real-world tabular datasets for classification and regression and 7 baseline methods. To be specific about our models, `Naïve-STaSy` is a naive conversion of SGMs without our proposed training strategies, and `STaSy` is trained with self-paced learning and the fine-tuning method. ii) In general, we follow the "train on synthetic, test on real (TSTR)" framework (Esteban et al., 2017; Jordon et al., 2019), which is a widely used evaluation method for tabular data (Xu et al., 2019; Kim et al., 2021; Lee et al., 2021), to evaluate the quality of sampling — in other words, we train various models, including DecisionTree, AdaBoost, Logistic/Linear Regression, MLP classifier/regressor, RandomForest, and XGBoost, with fake data, validate with original training data, and test them with real test data. For `Identity`, we train with real training data, choose the best performing model using the cross-validation, and test with real test data, whose score can be a criterion to evaluate the sampling quality of various generative methods. iii) We use various metrics to evaluate in various aspects. For the sampling quality, we mainly use average F1 for classification, and also report AUROC and Weighted-F1. We use $R^2$ and RMSE for regression. For the sampling diversity, we use coverage (Naeem et al., 2020), which was proposed to measure the diversity of generated records. Full results are in Appendix G. Detailed environments and hyperparameter settings are in Appendix H and I, respectively.

### 4.2 EXPERIMENTAL RESULTS

#### 4.2.1 SAMPLING QUALITY

Table 2 summarizes the key results on the sampling quality. We use task-oriented metrics, such as F1, $R^2$, and so on, under the TSTR evaluation framework. `MedGAN` and `VEEGAN`, two early GAN-based methods, show relatively lower test scores than other GAN-based methods, i.e., `CTGAN`, `TableGAN`, and `OCT-GAN`. In general, `CTGAN` and `OCT-GAN` show reliable quality among the baselines. However, our two score-based models always mark the best and the second best quality.

Our methods, `Naïve-STaSy` and `STaSy`, significantly outperform all the baselines by large margins. In particular, our methods perform well in small datasets, e.g., `Crowdsource`, `Obesity`, and `Robot`, while other methods show poor quality, as shown in Table 14 of Appendix G.1. These multi-class classification datasets have a small number of per class record, e.g., the smallest class in `Crowdsource` has 79 records in the training set, which means that our methods are able to capture fine-grained modes from the original data. Moreover, in `Credit`, which has a severe class imbalance ratio of 99.7% for class 0 and 0.3% for class 1, more than half of the baselines failed to generate the minority class, showing an F1 score close to 0 in Table 11 of Appendix G.1. `CTGAN` and `TableGAN` also achieve a good F1 score close to that of `Identity`, but `STaSy` takes the first place again.

Table 2: Classification/regression with real data. We report the average F1 (resp. macro F1), AUROC, and Weighted-F1 for binary (resp. multi-class) classification, and $R^2$ and RMSE for regression. The best (resp. the second best) results are highlighted in bold face (resp. with underline).

| Methods | Classification | | | Regression | |
|---|---|---|---|---|---|
| | F1 | AUROC | Weighted-F1 | $R^2$ | RMSE |
| Identity | 0.796 | 0.932 | 0.801 | 0.452 | 0.772 |
| MedGAN | 0.232 | 0.608 | 0.221 | -6.885 | 2.589 |
| VEEGAN | 0.329 | 0.661 | 0.316 | -4.901 | 2.500 |
| CTGAN | 0.627 | 0.838 | 0.635 | 0.122 | 0.946 |
| TVAE | 0.539 | 0.795 | 0.542 | 0.056 | 0.959 |
| TableGAN | 0.567 | 0.826 | 0.572 | -0.788 | 1.227 |
| OCT-GAN | 0.622 | 0.860 | 0.626 | 0.104 | 0.946 |
| RNODE | 0.496 | 0.796 | 0.502 | -0.483 | 1.112 |
| Naïve-STaSy | 0.768 | 0.917 | 0.776 | 0.321 | 0.789 |
| STaSy | **0.783** | **0.922** | **0.790** | **0.361** | **0.766** |

As flow-based generative models and SGMs with the *probability flow* method can calculate the exact log-probability of records, we present the log-probability as another metric for the sampling quality. Table 3 shows the median of the log-probabilities of testing records, averaged over all datasets. Since the log-probability is not bounded, we take a median of them to handle the case of outliers. Our methods, even without the

Table 3: The median of the log-probabilities of testing records, averaged over all datasets

| Methods | Log-probability |
|---|---|
| RNODE | 59.327 |
| STaSy w/o fine-tuning | 129.293 |
| STaSy | **131.734** |

fine-tuning, show a much better log-probability than RNODE, which optimizes the log-probability as its objective. Moreover, the median log-probability of testing records even improves after the proposed fine-tuning method.

Putting it all together, our proposed score-based generative models, i.e., Naïve-STaSy and STaSy, show reasonable performance in all cases regarding the machine learning efficacy and log-probability. Furthermore, as shown in Tables 2 and 3, the sampling quality always improves with the proposed training strategies, i.e., the self-paced learning and the fine-tuning, which justifies their efficacy.

### 4.2.2 SAMPLING DIVERSITY

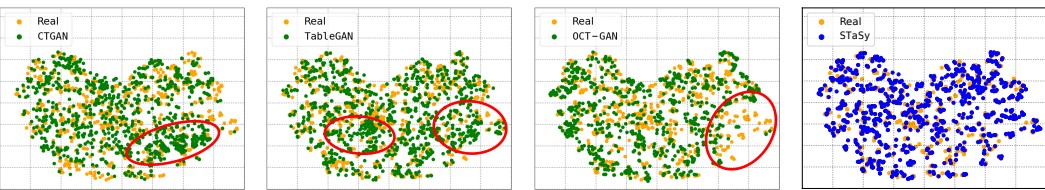

Figure 2: t-SNE visualizations of fake and the original records in Robot.

For the quantitative evaluation of the sampling diversity between existing methods and our proposed method, we use the coverage score (Naeem et al., 2020), which is bounded between 0 and 1. Coverage is the ratio of real records that have at least one fake record in its manifold. A manifold is a sphere around the sample with radius $r$, where $r$ is the distance between the sample and the $k$-th nearest neighborhood. Table 4 summarizes the averaged coverage of each method. MedGAN and VEEGAN show poor coverage scores, close to 0. This trend is also shown in the t-SNE visualizations in Appendix J.2. Among the baseline methods, CTGAN performs the best in terms of the sampling quality, whereas it

Table 4: Sampling diversity in terms of coverage averaged over all datasets

| Methods | Coverage |
|---|---|
| MedGAN | 0.037 |
| VEEGAN | 0.038 |
| CTGAN | 0.352 |
| TVAE | 0.494 |
| TableGAN | 0.434 |
| OCT-GAN | 0.381 |
| RNODE | 0.328 |
| Naïve-STaSy | 0.637 |
| STaSy w/o fine-tuning | 0.655 |
| STaSy | **0.658** |

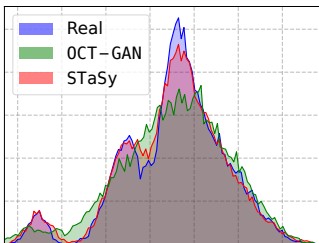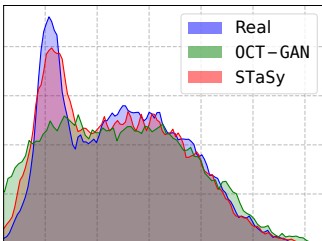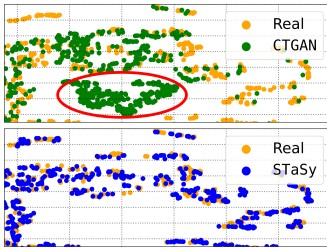

Figure 3: (Left and Middle) Histograms of values in *Roundness* and *Compactness* columns of `Bean`, respectively. (Right) t-SNE (van der Maaten & Hinton, 2008) visualizations of the fake and original records in `Obesity`. More visualizations are in Appendix J.

shows relatively inferior coverage performance than others. In specific, in `Robot`, `STaSy` shows a coverage of 0.94, while other three top-performing baselines, `CTGAN`, `TableGAN`, and `OCT-GAN`, show coverage scores less than 0.26 in Table 20 of Appendix G.2. Figure 2 also presents the diversity of each fake data by each method qualitatively, which reflects the results of coverage. In general, `STaSy` shows stable performance across the sampling quality and the sampling diversity, outperforming others by large margins.

In Figure 3 (Left and Middle), the fake data by `STaSy` shows an almost identical distribution to that of real data. In contrast, `OCT-GAN`, which was proposed to address the multi-modality issue of tabular data, fails to do it. This means `STaSy` is able to capture every mode in the columns, while `OCT-GAN` is not. In Figure 3 (Right), `CTGAN` generates some out-of-distribution records, highlighted in red.

### 4.2.3 SAMPLING TIME

Table 5: Runtime evaluation results, averaged over all datasets

| Methods | MedGAN | VEEGAN | CTGAN | TVAE | TableGAN | OCT-GAN | RNODE | Naïve-STaSy | STaSy |
|---|---|---|---|---|---|---|---|---|---|
| Runtime | 0.246 | 0.109 | 0.704 | 0.100 | **0.046** | 26.926 | 13.392 | 8.855 | 10.663 |

We summarize runtime in Table 5. In order to compare the runtime of all methods, we measure the wall-clock time taken to sample $N$ records, where $N$ is training size, 5 times, and average them. In general, simple GAN-based methods, especially `TableGAN` and `TVAE`, show faster runtime. On the other hand, SGMs, `OCT-GAN`, and `RNODE` take a relatively long time for sampling. Our proposed methods, `Naïve-STaSy` and `STaSy`, take a long sampling time compared to simple GAN-based methods but are faster than `OCT-GAN` and `RNODE`, which means a well-balanced trade-off between the sampling quality, diversity, and time.

### 4.3 ABLATION & SENSITIVITY STUDIES

Table 6: Ablation study. We report F1 (resp. $R^2$) for classification (resp. regression).

| Datasets | Naïve-STaSy | w/o fine-tuning | w/o SPL | STaSy |
|---|---|---|---|---|
| Credit | 0.737±0.083 | 0.752±0.053 | 0.775±0.050 | **0.795±0.034** |
| Default | 0.506±0.013 | 0.515±0.008 | 0.509±0.013 | **0.519±0.009** |
| Shoppers | 0.630±0.015 | 0.639±0.008 | 0.634±0.018 | **0.640±0.008** |
| Contraceptive | 0.393±0.043 | 0.424±0.014 | 0.406±0.016 | **0.425±0.003** |
| Crowdsource | 0.702±0.113 | 0.712±0.140 | 0.714±0.112 | **0.717±0.126** |
| Shuttle | 0.754±0.044 | 0.763±0.074 | 0.774±0.036 | **0.800±0.043** |
| Beijing | 0.648±0.088 | 0.672±0.108 | 0.648±0.088 | **0.679±0.115** |

We define three ablation models: 'Naïve-STaSy' without SPL and fine-tuning, 'w/o fine-tuning' without fine-tuning but with SPL, and 'w/o SPL' without SPL but with fine-tuning. All ablation

Table 7: Sensitivity analyses. We report F1 (resp. $R^2$) for classification (resp. regression).

| Datasets | SDE Type | Metric | $\alpha_0$ | Metric | $\beta_0$ | Metric |
|---|---|---|---|---|---|---|
| `Spambase` | VE | 0.875±0.023 | 0.05 | 0.875±0.023 | 0.70 | 0.907±0.027 |
| | VP | 0.880±0.029 | 0.10 | 0.909±0.028 | 0.80 | 0.909±0.028 |
| | sub-VP | **0.912±0.024** | 0.30 | 0.911±0.030 | 0.90 | 0.906±0.035 |
| `Crowdsource` | VE | **0.704±0.141** | 0.05 | 0.686±0.116 | 0.75 | 0.662±0.129 |
| | VP | 0.697±0.111 | 0.10 | 0.692±0.112 | 0.80 | 0.675±0.138 |
| | sub-VP | 0.630±0.115 | 0.30 | 0.669±0.127 | 0.95 | 0.683±0.112 |
| `Beijing` | VE | **0.679±0.115** | 0.05 | 0.667±0.107 | 0.70 | 0.667±0.107 |
| | VP | 0.645±0.086 | 0.10 | 0.673±0.114 | 0.75 | 0.663±0.101 |
| | sub-VP | 0.563±0.036 | 0.30 | 0.664±0.106 | 0.95 | 0.675±0.111 |

models are inferior to `STaSy` in Table 6, showing the effectiveness of SPL and fine-tuning. In particular, SPL improves the sampling diversity as in Figure 4. `Naïve-STaSy` suffers from mild mode collapses, as highlighted in red.

Table 7 shows sensitivity analyses w.r.t. some important hyperparameters. In general, all settings show reasonable results, outperforming the baselines. We recommend 0.2 and 0.25 for $\alpha_0$ and 0.9 and 0.95 for $\beta_0$.

We can adopt a variety of methods to solve the reverse SDE process in Equation 2. Our method can generate fake records with the *predictor-corrector* framework (Pred. Corr.) or the *probability flow* (PF) method (Song et al., 2021). The former uses the ancestral sampling (AS), reverse diffusion (RD), or Euler-Maruyama (EM) method for solving the reverse SDE, and for the correction process, the Langevin corrector. In Table 8, the *probability flow* method in Equation 10 mostly leads to successful results, and other datasets also show similar results.

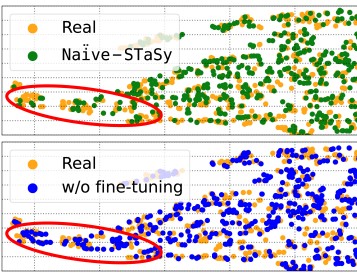

Figure 4: t-SNE visualizations of the fake and original records in `Beijing`

Table 8: Results of `Naïve-STaSy` by various reverse SDE solvers in `Magic`, where 'Pred.' means the predictor-only method. We report F1.

| Predictor | Pred. | Pred. Corr. |
|---|---|---|
| AS | 0.754±0.028 | 0.780±0.054 |
| RD | 0.777±0.059 | 0.779±0.066 |
| EM | 0.777±0.058 | 0.772±0.056 |
| PF | **0.781±0.058** | No Corr. |

## 5 CONCLUSIONS AND DISCUSSIONS

Synthesizing tabular data is an important yet non-trivial task, as it requires modeling a joint probability of multi-modal columns. To this end, we presented our detailed designs and experimental results with thorough analyses. Our proposed method, `STaSy`, is a score-based model equipped with our proposed self-paced learning and fine-tuning methods. In our experiments with 15 benchmark datasets and 7 baselines, `STaSy` outperforms other deep learning methods in terms of the sampling quality and diversity (and with an acceptable sampling time). Based on these considerations, we believe that `STaSy` shows significant advancements in tabular data synthesis. We expect much follow-up work in utilizing SGMs for tabular data synthesis.

**Limitations.** Although our model shows the best balance for the deep generative task trilemma, we think that there exists room to improve runtime further — existing simple GAN-based methods are faster than our method for sampling fake records. In addition, SGMs are known to be sometimes unstable for high-dimensional data, e.g., high-resolution images, but in general, stable for low-dimensional data, e.g., tabular data. Therefore, we think that SGMs have much potential for tabular data synthesis in the future.

## 6 ETHICS STATEMENT

Indeed, people do not always use artificial intelligence technology for righteous purposes. One can use our method to achieve his/her wrongful goals, e.g., selling high-quality fake data generated by our method, and retrieving private original data records from synthetic data. However, we believe that our research has much more beneficial points. One can use our method to generate fake data and share (after hiding the original data) to prevent potential privacy leakages. We, of course, need more studies to achieve the privacy protection goal based on our model. However, a research trend exists where researchers try to use a deep generative model to protect privacy (Park et al., 2018; Lee et al., 2021).

## 7 REPRODUCIBILITY STATEMENT

To reproduce the experimental results, we have made the following efforts: 1) Source codes used in the experiments are available in the supplementary material. By following the README guidance, the main results are easily reproducible. 2) All the experiments are repeated five times, and their mean and standard deviation values are reported in Appendix. 3) We provide extensive experimental details in Appendix H.

## ACKNOWLEDGMENTS

Jayoung Kim and Chaejeong Lee equally contributed. Noseong Park is the corresponding author. This work was supported by the Institute of Information & communications Technology Planning Evaluation (IITP) grant funded by the Korea government (MSIT) (90% from No. 2021-0-00231, Development of Approximate DBMS Query Technology to Facilitate Fast Query Processing for Exploratory Data Analysis and 10% from No. 2020-0-01361, Artificial Intelligence Graduate School Program (Yonsei University)).

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

## A  VE, VP, AND SUB-VP SDEs

We introduce the definitions of $f$ and $g$ as follows:

$$f(t) = \begin{cases} 0, & \text{if VE,} \\ -\frac{1}{2}\gamma(t)\mathbf{x}, & \text{if VP,} \\ -\frac{1}{2}\gamma(t)\mathbf{x}, & \text{if sub-VP,} \end{cases} \tag{11}$$

$$g(t) = \begin{cases} \sqrt{\frac{d[\sigma^2(t)]}{dt}}, & \text{if VE,} \\ \sqrt{\gamma(t)}, & \text{if VP,} \\ \sqrt{\gamma(t)(1 - e^{-2\int_0^t \gamma(s)\,ds})}, & \text{if sub-VP,} \end{cases} \tag{12}$$

where $\sigma(t)$ and $\gamma(t)$ are noise functions w.r.t. time $t$. $\sigma(t) = \sigma_{min}\left(\frac{\sigma_{max}}{\sigma_{min}}\right)^t$ for $t \in [0, 1]$, where $\sigma_{min}$ and $\sigma_{max}$ are hyperparameters, and we use $\sigma_{min} = \{0.01, 0.1\}$ and $\sigma_{max} = \{5.0, 10.0\}$. $\gamma(t) = \gamma_{min} + t\,(\gamma_{max} - \gamma_{min})$ for $t \in [0, 1]$, where $\gamma_{min}$ and $\gamma_{max}$ are hyperparameters, and we use $\gamma_{min} = \{0.01, 0.1\}$ and $\gamma_{max} = \{5.0, 10.0\}$.

## B  COMPARISON BETWEEN SOS AND STASY FOR THE OVERSAMPLING TASK

Table 9: Comparison between `SOS` and `STaSy` in terms of Weighted-F1

| Methods | Default | HTRU | Magic | Shoppers | Robot |
|---|---|---|---|---|---|
| Identity | 0.5146±0.0072 | 0.8657±0.0163 | 0.7807±0.0348 | 0.5673±0.0058 | 0.9031±0.0848 |
| SOS | 0.5618±0.0016 | 0.8767±0.0017 | 0.7949±0.0011 | 0.6568±0.0055 | 0.9197±0.0040 |
| STaSy | **0.5683±0.0026** | **0.8803±0.0027** | **0.7960±0.0015** | **0.6595±0.0044** | **0.9270±0.0029** |

In this section, we discuss the difference between `SOS` and `STaSy`. They are both based on SGMs, but they are optimized towards different goals by using different objective functions and training strategies. `SOS` has many design points specialized to augment minor classes only (rather than synthesizing entire tabular data) — for instance, `SOS` adopts a style-transfer-based idea to convert a major class sample to a minor one via their own SGM model without any consideration on the training difficulty of the denoising score matching. However, our `STaSy` more focuses on synthesizing entire tabular data by proposing special self-paced training and fine-tuning methods.

We use five imbalanced datasets to oversample — many datasets in our main experiments are not imbalanced. We conduct the oversampling experiments with `STaSy` to compare with `SOS` (Kim et al., 2022). In this experiment, `STaSy` is converted to an oversampling method following the design guidance of `SOS`, i.e., each minority class has its own score network and is separately trained. We train, for fair comparison, `STaSy` w/o fine-tuning for each minor class and generate minority samples to be the same size of the majority class. We compare the two models in terms of the sampling quality using Weighted-F1 which is specialized in evaluating imbalanced data. We note that `Identity` means that we do not use any oversampling methods, which is, therefore, a minimum quality requirement upon oversampling.

As shown in Table 9, `STaSy` w/o fine-tuning outperforms `SOS`. The result shows that our proposed training strategy, i.e., the self-paced learning, improves the model training regardless of tasks.

## C  NETWORK ARCHITECTURE

We propose the following score network $S_{\boldsymbol{\theta}}(\mathbf{x}(t), t)$:

$$\mathbf{h}_0 = \mathbf{x}(t),$$
$$\mathbf{h}_i = \omega(\mathtt{H}_i(\mathbf{h}_{i-1}, t) \oplus \mathbf{h}_{i-1}), 1 \le i \le d_N$$
$$S_{\boldsymbol{\theta}}(\mathbf{x}(t), t) = \mathtt{FC}(\mathbf{h}_{d_N}),$$

where $\mathbf{x}(t)$ is a record (or a row) at time $t$ in tabular data, $\mathbf{h}_i$ is the $i$-th hidden vector, and $\omega$ is an activation function. $d_N$ is the number of hidden layers. For various layer types of $\mathrm{H}_i(\mathbf{h}_{i-1}, t)$, we provide the following options:

$$
\mathrm{H}_i(\mathbf{h}_{i-1}, t) = \begin{cases} \mathrm{FC}_i(\mathbf{h}_{i-1}) \odot \psi(\mathrm{FC}_i^t(t)), & \text{if Squash,} \\ \mathrm{FC}_i(t \oplus \mathbf{h}_{i-1}), & \text{if Concat,} \\ \mathrm{FC}_i(\mathbf{h}_{i-1}) \odot \psi(\mathrm{FC}_i^{gate}(t) + \mathrm{FC}_i^{bias}(t)), & \text{if Concatsquash,} \end{cases}
$$

where we can choose one of the three possible layer types as a hyperparameter, $\odot$ means the element-wise multiplication, $\oplus$ means the concatenation operator, $\psi$ is the Sigmoid function, and $\mathrm{FC}$ is a fully connected layer. We modify the architecture of (Song et al., 2021) by using the layer types, being inspired by (Grathwohl et al., 2018).

## D  THRESHOLD CONTROLLING MECHANISM

Our threshold controlling mechanism is designed to meet the following 3 requirements: i) it can control when the entire dataset is used for training, starting from a subset, ii) it should gradually increase the size of used training records while logarithmically decreasing the number of hard records (that are not involved in training), and iii) it should be a monotonically increasing/decreasing function to guarantee that the training difficulty gets more challenging as the training process goes on.

The threshold controlling variables $\alpha$ and $\beta$, where $0 \leq \alpha \leq \beta \leq 1$, are gradually increased to 1 to involve the entire data records for training. We increase them proportionally to training steps, where $\alpha = \alpha_0 + \log\left(1 + c\left(\frac{e-1}{S}\right)(1 - \alpha_0)\right)$ and $\beta = \beta_0 + \log\left(1 + c\left(\frac{e-1}{S}\right)(1 - \beta_0)\right)$. $e$ is the base of the natural logarithm, $\alpha_0$ and $\beta_0$ are initial values of $\alpha$ and $\beta$, $c$ is the current training step, and $S$ determines when to utilize the entire data records. We use 10,000 for $S$. We set $\beta_0$ at least 0.8 to ensure 80% of the data records are involved in training at the start of the training. We set $\alpha = \alpha_0$ and $\beta = \beta_0$ at the beginning of the training, since $c$ is 0. We note that a training sample $x_i$ whose quantile of loss value is greater than $\beta$ is regarded as a hard sample and $v_i$ is set to be 0. If we set $\beta_0$ to be 0.8, the top 20% of difficult samples will not be used at the first training step. In this way, one can control the proportion of training samples involved in the training from the beginning.

## E  PROOF OF THEOREM 1

As defined in Section 3.2, the $\mathtt{STaSy}$ objective is as follows:

$$
\min_{\boldsymbol{\theta}, \mathbf{v}} \sum_{i=1}^N v_i l_i - \frac{Q(\alpha) - Q(\beta)}{2} \sum_{i=1}^N v_i^2 - Q(\beta) \sum_{i=1}^N v_i, \tag{13}
$$

where $l_i$ is the score matching loss for $i$-th training record as in Equation 6. We can rewrite the optimal solution for each training record $v_i$ with respect to fixed $\boldsymbol{\theta}$ in the vertex form. Let $\mathcal{L}(v_i)$ be the objective with fixed $\boldsymbol{\theta}$, which is a quadratic function with respect to $v_i$. Then,

$$
\begin{aligned}
\mathcal{L}(v_i) &= v_i l_i - \frac{Q(\alpha) - Q(\beta)}{2} v_i^2 - Q(\beta) v_i \\
&= -\frac{Q(\alpha) - Q(\beta)}{2} v_i^2 + (l_i - Q(\beta)) v_i \\
&= -\frac{Q(\alpha) - Q(\beta)}{2}\left(v_i^2 - \frac{2(l_i - Q(\beta))}{Q(\alpha) - Q(\beta)} v_i\right) \\
&= -\frac{Q(\alpha) - Q(\beta)}{2}\left\{v_i^2 - \frac{2(l_i - Q(\beta))}{Q(\alpha) - Q(\beta)} v_i + \left(\frac{l_i - Q(\beta)}{Q(\alpha) - Q(\beta)}\right)^2 - \left(\frac{l_i - Q(\beta)}{Q(\alpha) - Q(\beta)}\right)^2\right\} \\
&= -\frac{Q(\alpha) - Q(\beta)}{2}\left(v_i - \frac{l_i - Q(\beta)}{Q(\alpha) - Q(\beta)}\right)^2 + \frac{Q(\alpha) - Q(\beta)}{2}\left(\frac{l_i - Q(\beta)}{Q(\alpha) - Q(\beta)}\right)^2.
\end{aligned} \tag{14}
$$

Because $\frac{Q(\alpha)-Q(\beta)}{2}$ is less than or equal to 0 and $\frac{Q(\alpha)-Q(\beta)}{2}\left(\frac{l_i-Q(\beta)}{Q(\alpha)-Q(\beta)}\right)^2$ is a constant, the solution $v_i$ which minimizes Equation 14 is $v_i = \frac{l_i-Q(\beta)}{Q(\alpha)-Q(\beta)}$. Considering $v_i \in [0, 1]$, we can get the optimal $v_i$ as follows:

$$v_i^* = \begin{cases} 1, & \text{if } l_i \leq Q(\alpha), \\ 0, & \text{if } l_i \geq Q(\beta), \\ \dfrac{l_i - Q(\beta)}{Q(\alpha) - Q(\beta)}, & \text{otherwise.} \end{cases} \tag{15}$$

## F   THE HUTCHINSON'S ESTIMATION AS A FINE-TUNING OBJECTIVE

We use the denoising score matching loss in Line 10 of Algorithm 1. In this section, we describe the results of an additional experiment in which the Hutchinson's log-probability estimation is used for the fine-tuning objective.

Table 10 summarizes the F1 score and the median of log-probabilities when we use the denoising score matching loss and the Hutchinson's estimation as the tine-tuning objective. In `Contraceptive`, there does not exist a clear winner between the two fine-tuning objectives, and similar results are also shown in other datasets. However, in some datasets, e.g., `Shoppers` and `Crowdsource`, the former shows better F1 scores and better medians of the log-probabilities than the latter by large margins. In addition, when we update $\theta$ using the Hutchinson's estimation, in `Default`, the sampling quality is lower than before fine-tuning. Considering these results, we use the denoising score matching loss, which shows the generalizability, as our default fine-tuning objective.

Table 10: We report the F1 score and the median of the log-probabilities of testing records according to the fine-tuning objective.

| Datasets | STaSy w/o fine-tuning | | Fine-tine with the denoising score matching loss | | Fine-tune with the Hutchinson's estimation | |
|---|---|---|---|---|---|---|
| | F1 | Log-probability | F1 | Log-probability | F1 | Log-probability |
| Default | 0.515±0.008 | 131.011 | **0.519±0.009** | **131.122** | 0.512±0.008 | 106.184 |
| Shoppers | 0.639±0.008 | 189.422 | **0.640±0.008** | **220.589** | 0.636±0.009 | 206.924 |
| Contraceptive | 0.424±0.014 | 225.334 | **0.425±0.003** | **225.638** | 0.425±0.011 | 225.392 |
| Crowdsource | 0.712±0.140 | 47.567 | **0.717±0.126** | **48.905** | 0.710±0.135 | 48.683 |

## G   ADDITIONAL EXPERIMENTAL RESULTS

### G.1   SAMPLING QUALITY

We mainly use F1 (resp. $R^2$) for the classfication (resp. regression) TSTR evaluation, and also report AUROC and Weighted-F1 (resp. RMSE) results. Full results for all datasets are in Tables 11, 12, and 13 for binary classification, Tables 14, 15, and 16 for multi-class classification, and Table 17 for regression. We train and test various base classifiers/regressors and report their mean and standard deviation. Moreover, we use the log-probability as another metric for the sampling quality. Full results are in Table 18. The best results are highlighted in bold face and the second best results with underline. As shown, `Naïve-STaSy` and `STaSy` show the best and the second best performances in almost all cases.

Table 11: Classification with real data. We report F1 for binary classification.

| Methods | Binary classification | | | | | | |
| --- | --- | --- | --- | --- | --- | --- | --- |
| | Credit | Default | HTRU | Magic | Phishing | Shoppers | Spambase |
| Identity | 0.765±0.071 | 0.442±0.039 | 0.880±0.009 | 0.778±0.064 | 0.948±0.024 | 0.601±0.059 | 0.934±0.055 |
| MedGAN | 0.000±0.000 | 0.000±0.000 | 0.000±0.000 | 0.553±0.068 | 0.615±0.000 | 0.198±0.132 | 0.438±0.217 |
| VEEGAN | 0.000±0.000 | 0.367±0.002 | 0.403±0.256 | 0.546±0.063 | 0.795±0.108 | 0.322±0.082 | 0.522±0.091 |
| CTGAN | 0.743±0.070 | 0.489±0.024 | 0.851±0.008 | 0.726±0.032 | 0.894±0.008 | 0.509±0.052 | 0.773±0.030 |
| TVAE | 0.026±0.044 | 0.432±0.038 | 0.849±0.008 | 0.693±0.016 | 0.911±0.006 | 0.573±0.030 | 0.752±0.036 |
| TableGAN | 0.729±0.040 | 0.425±0.009 | 0.841±0.005 | 0.736±0.048 | 0.903±0.007 | 0.586±0.062 | 0.710±0.128 |
| OCT-GAN | 0.113±0.171 | 0.484±0.016 | 0.863±0.011 | 0.725±0.018 | 0.904±0.006 | 0.619±0.037 | 0.858±0.023 |
| RNODE | 0.077±0.082 | 0.395±0.013 | 0.582±0.035 | 0.746±0.032 | 0.894±0.013 | 0.554±0.026 | 0.822±0.048 |
| Naïve-STaSy | 0.737±0.083 | 0.506±0.013 | 0.881±0.005 | 0.781±0.058 | 0.925±0.010 | 0.630±0.015 | 0.897±0.031 |
| STaSy | **0.795±0.034** | **0.519±0.009** | **0.882±0.005** | **0.783±0.058** | **0.930±0.012** | **0.640±0.008** | **0.912±0.024** |

Table 12: Classification with real data. We report AUROC for binary classification.

| Methods | Binary classification | | | | | | |
| --- | --- | --- | --- | --- | --- | --- | --- |
| | Credit | Default | HTRU | Magic | Phishing | Shoppers | Spambase |
| Identity | 0.968±0.021 | 0.764±0.016 | 0.969±0.005 | 0.907±0.039 | 0.988±0.009 | 0.920±0.014 | 0.983±0.018 |
| MedGAN | 0.500±0.000 | 0.500±0.000 | 0.470±0.047 | 0.688±0.070 | 0.660±0.174 | 0.764±0.067 | 0.809±0.067 |
| VEEGAN | 0.660±0.109 | 0.498±0.061 | 0.818±0.192 | 0.773±0.036 | 0.869±0.121 | 0.782±0.075 | 0.707±0.066 |
| CTGAN | 0.957±0.033 | 0.749±0.006 | 0.950±0.020 | 0.869±0.025 | 0.969±0.007 | 0.847±0.031 | 0.904±0.027 |
| TVAE | 0.702±0.120 | 0.743±0.012 | 0.954±0.012 | 0.843±0.029 | 0.979±0.004 | 0.861±0.020 | 0.901±0.031 |
| TableGAN | 0.928±0.051 | 0.660±0.028 | 0.957±0.013 | 0.887±0.027 | 0.972±0.006 | 0.856±0.038 | 0.929±0.041 |
| OCT-GAN | 0.803±0.114 | 0.726±0.012 | 0.957±0.015 | 0.865±0.018 | 0.973±0.005 | 0.888±0.017 | 0.949±0.020 |
| RNODE | 0.882±0.093 | 0.718±0.013 | 0.928±0.059 | 0.874±0.032 | 0.966±0.007 | 0.890±0.021 | 0.931±0.046 |
| Naïve-STaSy | 0.958±0.038 | **0.749±0.018** | **0.967±0.011** | **0.899±0.039** | 0.984±0.005 | 0.908±0.009 | 0.969±0.025 |
| STaSy | **0.959±0.039** | 0.744±0.019 | 0.966±0.009 | 0.899±0.038 | **0.986±0.005** | **0.915±0.008** | **0.976±0.019** |

Table 13: Classification with real data. We report Weighted-F1, which is inversely weighted to its class size, for binary classification.

| Methods | Binary classification | | | | | | |
| --- | --- | --- | --- | --- | --- | --- | --- |
| | Credit | Default | HTRU | Magic | Phishing | Shoppers | Spambase |
| Identity | 0.770±0.073 | 0.541±0.030 | 0.890±0.009 | 0.818±0.051 | 0.953±0.022 | 0.650±0.051 | 0.944±0.021 |
| MedGAN | 0.002±0.000 | 0.197±0.000 | 0.087±0.000 | 0.482±0.166 | 0.343±0.003 | 0.288±0.098 | 0.527±0.121 |
| VEEGAN | 0.002±0.000 | 0.285±0.000 | 0.419±0.258 | 0.520±0.106 | 0.799±0.115 | 0.424±0.039 | 0.570±0.083 |
| CTGAN | 0.746±0.069 | 0.571±0.019 | 0.864±0.007 | 0.775±0.025 | 0.906±0.007 | 0.560±0.048 | 0.802±0.028 |
| TVAE | 0.041±0.061 | 0.531±0.029 | 0.862±0.007 | 0.735±0.009 | 0.919±0.006 | 0.624±0.028 | 0.770±0.034 |
| TableGAN | 0.737±0.037 | 0.506±0.012 | 0.855±0.005 | 0.786±0.038 | 0.913±0.006 | 0.642±0.057 | 0.771±0.093 |
| OCT-GAN | 0.103±0.160 | 0.560±0.019 | 0.874±0.010 | 0.770±0.015 | 0.912±0.005 | 0.664±0.032 | 0.877±0.021 |
| RNODE | 0.089±0.080 | 0.503±0.010 | 0.620±0.037 | 0.789±0.027 | 0.904±0.011 | 0.610±0.023 | 0.853±0.035 |
| Naïve-STaSy | 0.744±0.085 | 0.590±0.010 | **0.892±0.006** | 0.820±0.047 | 0.932±0.009 | 0.673±0.012 | 0.911±0.028 |
| STaSy | **0.795±0.036** | **0.596±0.008** | 0.891±0.005 | **0.820±0.046** | **0.937±0.011** | **0.681±0.006** | **0.926±0.020** |

Table 14: Classification with real data. We report macro F1 for multi-class classification.

| Methods | Multi-class classification | | | | | |
|---|---|---|---|---|---|---|
| | Bean | Contraceptive | Crowdsource | Obesity | Robot | Shuttle |
| Identity | 0.932±0.009 | 0.470±0.023 | 0.760±0.100 | 0.964±0.010 | 0.973±0.037 | 0.904±0.110 |
| MedGAN | 0.058±0.000 | 0.412±0.017 | 0.153±0.013 | 0.187±0.082 | 0.281±0.063 | 0.126±0.000 |
| VEEGAN | 0.270±0.095 | 0.313±0.032 | 0.137±0.000 | 0.176±0.045 | 0.297±0.008 | 0.126±0.000 |
| CTGAN | 0.881±0.014 | 0.340±0.016 | 0.525±0.120 | 0.116±0.018 | 0.646±0.037 | 0.655±0.143 |
| TVAE | 0.892±0.011 | 0.387±0.027 | 0.137±0.000 | 0.456±0.031 | 0.771±0.039 | 0.126±0.000 |
| TableGAN | 0.606±0.030 | 0.337±0.021 | 0.406±0.088 | 0.294±0.024 | 0.400±0.032 | 0.396±0.010 |
| OCT-GAN | 0.912±0.018 | 0.376±0.007 | 0.579±0.111 | 0.253±0.030 | 0.768±0.035 | 0.626±0.081 |
| RNODE | 0.805±0.114 | 0.350±0.012 | 0.137±0.000 | 0.472±0.064 | 0.494±0.112 | 0.126±0.000 |
| Naïve-STaSy | 0.930±0.010 | 0.393±0.043 | 0.702±0.114 | 0.902±0.026 | 0.940±0.033 | 0.754±0.044 |
| STaSy | **0.932±0.009** | **0.425±0.003** | **0.717±0.126** | **0.902±0.025** | **0.940±0.028** | **0.800±0.043** |

Table 15: Classification with real data. We report AUROC for multi-class classification.

| Methods | Multi-class classification | | | | | |
|---|---|---|---|---|---|---|
| | Bean | Contraceptive | Crowdsource | Obesity | Robot | Shuttle |
| Identity | 0.992±0.006 | 0.692±0.015 | 0.946±0.053 | 0.997±0.003 | 0.995±0.006 | 0.999±0.002 |
| MedGAN | 0.500±0.000 | 0.622±0.022 | 0.599±0.090 | 0.694±0.110 | 0.600±0.068 | 0.500±0.000 |
| VEEGAN | 0.720±0.115 | 0.602±0.045 | 0.500±0.000 | 0.574±0.020 | 0.585±0.054 | 0.500±0.000 |
| CTGAN | 0.972±0.036 | 0.515±0.024 | 0.881±0.045 | 0.496±0.008 | 0.820±0.037 | 0.963±0.044 |
| TVAE | 0.980±0.016 | 0.600±0.048 | 0.500±0.000 | 0.833±0.007 | 0.939±0.045 | 0.500±0.000 |
| TableGAN | 0.898±0.040 | 0.563±0.013 | 0.872±0.058 | 0.737±0.034 | 0.701±0.047 | 0.784±0.076 |
| OCT-GAN | 0.982±0.021 | 0.581±0.018 | 0.913±0.053 | 0.663±0.038 | 0.923±0.026 | 0.960±0.046 |
| RNODE | 0.973±0.026 | 0.532±0.020 | 0.500±0.000 | 0.818±0.072 | 0.842±0.087 | 0.500±0.000 |
| Naïve-STaSy | 0.986±0.017 | 0.631±0.016 | 0.931±0.072 | 0.981±0.025 | **0.988±0.012** | 0.967±0.058 |
| STaSy | **0.991±0.008** | **0.651±0.004** | **0.950±0.046** | **0.987±0.012** | 0.987±0.014 | **0.980±0.036** |

Table 16: Classification with real data. We report Weighted-F1, which is inversely weighted to its class size, for multi-class classification.

| Methods | Multi-class classification | | | | | |
|---|---|---|---|---|---|---|
| | Bean | Contraceptive | Crowdsource | Obesity | Robot | Shuttle |
| Identity | 0.935±0.009 | 0.448±0.017 | 0.630±0.276 | 0.962±0.015 | 0.967±0.049 | 0.902±0.109 |
| MedGAN | 0.050±0.000 | 0.385±0.012 | 0.066±0.007 | 0.180±0.074 | 0.230±0.052 | 0.032±0.000 |
| VEEGAN | 0.272±0.095 | 0.294±0.036 | 0.050±0.000 | 0.181±0.042 | 0.257±0.018 | 0.032±0.000 |
| CTGAN | 0.882±0.015 | 0.331±0.017 | 0.481±0.115 | 0.116±0.017 | 0.642±0.047 | 0.583±0.166 |
| TVAE | 0.893±0.011 | 0.369±0.021 | 0.050±0.000 | 0.453±0.021 | 0.768±0.040 | 0.032±0.000 |
| TableGAN | 0.606±0.031 | 0.325±0.022 | 0.337±0.096 | 0.292±0.020 | 0.363±0.033 | 0.308±0.016 |
| OCT-GAN | 0.915±0.016 | 0.364±0.009 | 0.533±0.120 | 0.251±0.027 | 0.764±0.036 | 0.558±0.108 |
| RNODE | 0.804±0.122 | 0.333±0.019 | 0.050±0.000 | 0.472±0.062 | 0.467±0.124 | 0.032±0.000 |
| Naïve-STaSy | 0.933±0.010 | 0.377±0.048 | 0.674±0.116 | 0.899±0.028 | 0.939±0.030 | 0.704±0.035 |
| STaSy | **0.935±0.009** | **0.400±0.004** | **0.678±0.153** | **0.903±0.025** | **0.939±0.026** | **0.769±0.079** |

Table 17: Regression with real data. We report $R^2$ and RMSE for regression.

| Methods | $R^2$ | | RMSE | |
|---|---|---|---|---|
| | Beijing | News | Beijing | News |
| Identity | 0.741±0.135 | 0.162±0.022 | 0.691±0.104 | 0.853±0.043 |
| MedGAN | -4.268±2.107 | -9.503±4.145 | 2.560±0.655 | 2.619±0.024 |
| VEEGAN | -6.440±0.566 | -3.362±0.702 | 3.121±0.098 | 1.880±0.162 |
| CTGAN | 0.210±0.217 | 0.035±0.032 | 1.012±0.131 | **0.880±0.012** |
| TVAE | 0.385±0.020 | -0.273±0.067 | 0.902±0.014 | 1.015±0.027 |
| TableGAN | 0.219±0.054 | -1.795±0.981 | 1.018±0.037 | 1.437±0.148 |
| OCT-GAN | 0.379±0.087 | -0.172±0.167 | 0.908±0.070 | 0.984±0.084 |
| RNODE | 0.282±0.117 | -1.248±1.237 | 0.974±0.078 | 1.251±0.270 |
| Naïve-STaSy | 0.648±0.088 | -0.006±0.088 | 0.678±0.082 | 0.900±0.039 |
| STaSy | **0.679±0.115** | **0.042±0.072** | **0.647±0.111** | 0.885±0.042 |

Table 18: The median of the log-probabilities of testing records are reported.

| | Datasets | RNODE | STaSy w/o fine-tuning | STaSy |
|---|---|---|---|---|
| Binary | Credit | 71.248 | 117.001 | **117.404** |
| | Default | 55.284 | 131.011 | **131.122** |
| | HTRU | **27.852** | 5.435 | 5.326 |
| | Magic | 17.413 | 30.247 | **30.216** |
| | Phishing | 53.478 | 228.912 | **228.914** |
| | Shoppers | 62.843 | 189.422 | **220.589** |
| | Spambase | 140.686 | 253.935 | **253.798** |
| Multi. | Bean | 46.764 | 92.555 | **92.635** |
| | Contraceptive | 26.797 | 225.334 | **225.638** |
| | Crowdsource | 23.323 | 47.567 | **48.905** |
| | Obesity | 130.018 | 330.003 | **333.017** |
| | Robot | 14.238 | 79.233 | **79.355** |
| | Shuttle | 48.875 | 114.247 | **114.294** |
| Reg. | Beijing | **42.614** | 17.720 | 18.192 |
| | News | **128.472** | 76.770 | 76.600 |

## G.2 SAMPLING DIVERSITY

We use the coverage score as a metric for the sampling diversity. Full results for all datasets are in Tables 19, 20, and 21. We measure the coverage score 5 times with different fake records and report their mean and standard deviation.

Coverage is bounded between 0 and 1, and higher coverage means more diverse samples. This k-NN-based measurement is expected to achieve 100% performance when the real and fake records are identical, but in practice, this is not always the case. For the dataset whose coverage score does not show 1 for two same data records, we choose the hyperparameter $k$ to achieve at least greater than 0.95. In our experiments, $k$ for `Phishing` is 7, and for others, $k$ is 5. As shown in Tables 19, 20, and 21, in 12 out of 15 datasets, our methods outperform others by large margins.

Table 19: Sampling diversity in terms of coverage for binary classification datasets

| Methods | Binary classification | | | | | | |
|---|---|---|---|---|---|---|---|
| | Credit | Default | HTRU | Magic | Phishing | Shoppers | Spambase |
| MedGAN | 0.000±0.000 | 0.000±0.000 | 0.000±0.000 | 0.001±0.000 | 0.002±0.001 | 0.000±0.000 | 0.002±0.000 |
| VEEGAN | 0.000±0.000 | 0.000±0.000 | 0.000±0.000 | 0.003±0.000 | 0.035±0.000 | 0.141±0.002 | 0.201±0.004 |
| CTGAN | 0.174±0.001 | 0.190±0.001 | 0.461±0.004 | 0.655±0.003 | 0.416±0.001 | 0.723±0.006 | 0.471±0.007 |
| TVAE | 0.373±0.001 | 0.272±0.002 | 0.741±0.001 | 0.650±0.003 | 0.623±0.002 | 0.737±0.001 | 0.698±0.002 |
| TableGAN | **0.458±0.002** | 0.216±0.001 | 0.299±0.002 | 0.748±0.002 | 0.512±0.007 | 0.698±0.005 | 0.711±0.005 |
| OCT-GAN | 0.000±0.000 | 0.171±0.000 | 0.375±0.004 | 0.634±0.001 | 0.450±0.003 | 0.714±0.003 | 0.470±0.016 |
| RNODE | 0.025±0.000 | **0.293±0.003** | 0.751±0.006 | 0.612±0.003 | 0.272±0.008 | 0.447±0.004 | 0.326±0.009 |
| Naïve-STaSy | 0.014±0.000 | 0.149±0.001 | **0.921±0.002** | 0.919±0.002 | 0.655±0.006 | **0.832±0.005** | 0.684±0.011 |
| STaSy w/o fine-tuning | 0.013±0.000 | 0.083±0.001 | 0.907±0.002 | 0.943±0.003 | **0.780±0.005** | 0.796±0.006 | **0.727±0.006** |
| STaSy | 0.014±0.000 | 0.101±0.001 | 0.911±0.003 | **0.944±0.001** | 0.779±0.005 | 0.798±0.004 | 0.727±0.013 |

Table 20: Sampling diversity in terms of coverage for multi-class classification datasets

| Methods | Multi-class classification | | | | | |
|---|---|---|---|---|---|---|
| | Bean | Contraceptive | Crowdsource | Obesity | Robot | Shuttle |
| MedGAN | 0.000±0.000 | 0.538±0.007 | 0.000±0.000 | 0.007±0.001 | 0.000±0.000 | 0.000±0.000 |
| VEEGAN | 0.004±0.001 | 0.082±0.002 | 0.000±0.000 | 0.100±0.003 | 0.005±0.001 | 0.001±0.000 |
| CTGAN | 0.053±0.001 | 0.753±0.011 | 0.064±0.000 | 0.252±0.015 | 0.140±0.004 | 0.031±0.000 |
| TVAE | 0.118±0.001 | 0.680±0.011 | 0.254±0.003 | 0.297±0.003 | 0.472±0.014 | 0.111±0.002 |
| TableGAN | 0.116±0.003 | 0.751±0.001 | 0.339±0.001 | 0.440±0.006 | 0.259±0.003 | 0.005±0.000 |
| OCT-GAN | 0.106±0.001 | 0.764±0.011 | 0.133±0.000 | 0.345±0.010 | 0.250±0.004 | 0.023±0.000 |
| RNODE | **0.292±0.006** | 0.547±0.011 | 0.104±0.003 | 0.375±0.008 | 0.113±0.004 | 0.005±0.000 |
| Naïve-STaSy | 0.092±0.003 | 0.839±0.012 | 0.920±0.004 | **0.825±0.006** | 0.935±0.007 | 0.133±0.003 |
| STaSy w/o fine-tuning | 0.095±0.005 | 0.879±0.007 | 0.970±0.002 | 0.777±0.013 | 0.937±0.004 | 0.207±0.001 |
| STaSy | 0.100±0.005 | **0.894±0.010** | **0.971±0.005** | 0.778±0.008 | **0.937±0.004** | **0.209±0.001** |

Table 21: Sampling diversity in terms of coverage for regression datasets

| Methods | Regression | |
|---|---|---|
| | Beijing | News |
| MedGAN | 0.000±0.000 | 0.000±0.000 |
| VEEGAN | 0.000±0.000 | 0.002±0.000 |
| CTGAN | 0.532±0.002 | 0.366±0.001 |
| TVAE | 0.720±0.002 | 0.665±0.001 |
| TableGAN | 0.803±0.003 | 0.154±0.003 |
| OCT-GAN | 0.693±0.000 | 0.582±0.000 |
| RNODE | 0.501±0.003 | 0.255±0.002 |
| Naïve-STaSy | 0.876±0.003 | 0.755±0.003 |
| STaSy w/o fine-tuning | **0.943±0.003** | 0.762±0.004 |
| STaSy | 0.941±0.003 | **0.762±0.002** |

### G.3 SAMPLING TIME

Tables 22, 23, and 24 show runtime evaluation results of each method. We measure the wall-clock time taken to sample fake records 5 times, and report their mean and standard deviation. In almost all datasets, `Naïve-STaSy` and `STaSy` show faster runtime than `OCT-GAN` and `RNODE`. `TableGAN` and `TVAE` take a short sampling time, but considering their inferior sampling quality and diversity, only our proposed model resolves the problems of *the generative learning trilemma*.

Table 22: Wall-clock runtime for binary classification datasets

| Methods | Binary classification | | | | | | |
|---|---|---|---|---|---|---|---|
| | Credit | Default | HTRU | Magic | Phishing | Shoppers | Spambase |
| MedGAN | 0.756±0.328 | 0.223±0.334 | 0.201±0.332 | 0.205±0.333 | 0.189±0.334 | 0.203±0.332 | 0.187±0.328 |
| VEEGAN | 0.800±0.031 | 0.112±0.008 | 0.051±0.005 | 0.061±0.007 | 0.041±0.009 | 0.041±0.006 | 0.019±0.002 |
| CTGAN | 6.342±0.300 | 0.546±0.054 | 0.299±0.085 | 0.284±0.002 | 0.163±0.005 | 0.204±0.005 | 0.107±0.008 |
| TVAE | 0.937±0.185 | 0.052±0.005 | 0.028±0.001 | 0.049±0.004 | 0.022±0.004 | 0.039±0.003 | 0.014±0.005 |
| TableGAN | **0.381±0.025** | **0.043±0.007** | **0.013±0.001** | **0.027±0.007** | **0.019±0.007** | **0.020±0.007** | **0.011±0.005** |
| OCT-GAN | 241.104±4.724 | 28.206±3.598 | 10.312±3.688 | 11.083±3.701 | 8.622±3.704 | 7.741±3.714 | 4.224 ±3.680 |
| RNODE | 10.613±0.139 | 12.901±0.731 | 4.346±0.073 | 2.576±0.622 | 11.385±0.067 | 16.461±1.326 | 21.270±0.121 |
| Naïve-STaSy | 79.202±0.350 | 7.192±0.320 | 3.275±0.245 | 3.466±0.254 | 0.733±0.239 | 3.774±0.327 | 2.234±0.229 |
| STaSy | 79.085±0.449 | 9.073±0.319 | 11.903±0.201 | 3.788±0.284 | 3.287±0.267 | 4.015±0.285 | 2.173±0.225 |

Table 23: Wall-clock runtime for multi-class classification datasets

| Methods | Multi-class classification | | | | | |
|---|---|---|---|---|---|---|
| | Bean | Contraceptive | Crowdsource | Obesity | Robot | Shuttle |
| MedGAN | 0.218±0.347 | 0.174±0.335 | 0.191±0.336 | 0.196±0.334 | 0.181±0.338 | 0.271±0.330 |
| VEEGAN | 0.029±0.006 | 0.006±0.000 | 0.033±0.006 | 0.009±0.001 | 0.024±0.010 | 0.102±0.004 |
| CTGAN | 0.227±0.009 | 0.025±0.002 | 0.252±0.027 | 0.036±0.002 | 0.113±0.014 | 0.712±0.009 |
| TVAE | **0.023±0.002** | 0.004±0.001 | 0.038±0.005 | 0.039±0.040 | 0.009±0.000 | 0.138±0.008 |
| TableGAN | 0.026±0.008 | **0.002±0.000** | **0.021±0.007** | **0.004±0.001** | **0.006±0.000** | **0.041±0.007** |
| OCT-GAN | 6.595±3.725 | 2.849±3.678 | 6.917±3.743 | 2.885±3.613 | 5.013±3.670 | 29.469±3.677 |
| RNODE | 5.210±0.234 | 12.452±0.270 | 7.985±0.072 | 23.675±0.432 | 10.861±0.801 | 6.819±0.257 |
| Naïve-STaSy | 2.797±0.263 | 1.346±0.220 | 3.090±0.237 | 1.387±0.232 | 1.692±0.215 | 12.935±0.853 |
| STaSy | 2.750±0.241 | 1.045±0.247 | 3.126±0.229 | 1.023±0.237 | 1.371±0.227 | 13.352±0.382 |

Table 24: Wall-clock runtime for regression datasets

| Methods | Regression | |
|---|---|---|
| | Beijing | News |
| MedGAN | 0.212±0.334 | 0.284±0.341 |
| VEEGAN | 0.067±0.005 | 0.234±0.037 |
| CTGAN | 0.346±0.023 | 0.902±0.013 |
| TVAE | 0.036±0.006 | 0.075±0.003 |
| TableGAN | **0.023±0.008** | **0.052±0.005** |
| OCT-GAN | 18.008±3.566 | 20.866±3.648 |
| RNODE | 19.380±0.500 | 34.955±0.276 |
| Naïve-STaSy | 7.338±0.146 | 2.362±0.241 |
| STaSy | 7.818±1.074 | 16.132±1.241 |

## H EXPERIMENTAL ENVIRONMENTS

Our software and hardware environments are as follows: UBUNTU 18.04 LTS, PYTHON 3.8.2, PYTORCH 1.8.1, CUDA 11.4, and NVIDIA Driver 470.42.01, i9 CPU, and NVIDIA RTX 3090. Our code for the experiments is mainly based on `https://github.com/yang-song/score_sde_pytorch` (Apache License 2.0).

## H.1 BASELINES

We utilize a set of baselines that includes various generative models.

- `Identity` is a case where we do not synthesize but use original data.
- `MedGAN`[1] (Choi et al., 2017) is a GAN that incorporates non-adversarial losses to generate discrete medical records.
- `VEEGAN`[1] (Srivastava et al., 2017) is a GAN for tabular data that avoids mode collapse by adding a reconstructor network.
- `CTGAN`[1] (Xu et al., 2019) and `TVAE`[1] (Xu et al., 2019) are a conditional GAN and a VAE for tabular data with mixed types of variables.
- `TableGAN`[1] (Park et al., 2018) is a GAN for tabular data using convolutional neural networks.
- `OCT-GAN`[2] (Kim et al., 2021) is a GAN that has a generator and discriminator based on neural ordinary differential equations.
- `RNODE`[3] (Finlay et al., 2020) is an advanced flow-based model with two regularization terms added to the training objective of FFJORD (Grathwohl et al., 2018).

## H.2 DATASETS

In this section, we describe 15 real-world tabular datasets for our experiments. We select the datasets for experiments with two metrics: 1) how many times a dataset has been cited/used in previous papers, and 2) how many times a dataset has been viewed/downloaded in famous repositories, such as UCI Machine Learning Repository and Kaggle. Among them, we choose the datasets that can be used for classification and regression tasks, with more than 5 columns and 1,000 rows.

- `Credit` is a binary classification dataset collected from European cardholders for credit card fraud detection.
- `Default` (Lichman, 2013) is a binary classification dataset describing the information on credit card clients in Taiwan regarding default payments.
- `HTRU` (Lyon, 2017) is a binary classification dataset that describes a sample of pulsar candidates collected during the High Time Resolution Universe Survey.
- `Magic` (Bock, 2007) is a binary classification dataset that simulates the registration of high-energy gamma particles in the atmospheric telescope.
- `Phishing` (Mohammad, 2015) is a binary classification dataset used to distinguish between phishing and legitimate web pages.
- `Shoppers` (C Okan Sakar, 2019) is a binary classification dataset about online shoppers' intention.
- `Spambase` (Hopkins, 1999) is a binary classification dataset that indicates whether an email is spam or non-spam.
- `Bean` (Koklu & Ozkan, 2020) is a multi-class classification dataset that includes types of beans with their characteristics.
- `Contraceptive` (Lim, 1997) is a multi-class classification dataset about Indonesia contraceptive prevalence.
- `Crowdsource` (Johnson & Iizuka, 2016) is a multi-class classification dataset used to classify satellite images into different land cover classes.
- `Obesity` (Palechor & de la Hoz Manotas, 2019) is a multi-class classification dataset describing obesity levels based on eating habits and physical condition.
- `Robot` (Freire, 2010) is a multi-class classification dataset collected as the robot moves around the room, following the wall using ultrasound sensors.

---

[1] `https://github.com/sdv-dev/SDGym` (MIT License)
[2] `https://github.com/bigdyl-yonsei/OCTGAN`
[3] `https://github.com/cfinlay/ffjord-rnode` (MIT License)

Table 25: Datasets used for our experiments

| Datasets | #train | #test | #continuous | #categorical | task (#class) |
|---|---|---|---|---|---|
| Credit | 264.8K | 20K | 29 | 1 | Binary classification |
| Default | 24K | 6K | 13 | 11 | Binary classification |
| HTRU | 14.3K | 3.6K | 8 | 1 | Binary classification |
| Magic | 15.2K | 3.8K | 10 | 1 | Binary classification |
| Phishing | 8.8K | 2.2K | 0 | 31 | Binary classification |
| Shoppers | 9.8K | 2.4K | 10 | 8 | Binary classification |
| Spambase | 3.7K | 0.9K | 57 | 1 | Binary classification |
| Bean | 10.8K | 2.7K | 16 | 1 | Multi-class classification (7) |
| Contraceptive | 1.2K | 0.3K | 0 | 10 | Multi-class classification (3) |
| Crowdsource | 8.6K | 2.1K | 28 | 1 | Multi-class classification (6) |
| Obesity | 1.6K | 0.4K | 7 | 10 | Multi-class classification (7) |
| Robot | 4.4K | 1.1K | 24 | 1 | Multi-class classification (4) |
| Shuttle | 46.4K | 11.6K | 9 | 1 | Multi-class classification (7) |
| Beijing | 15.2K | 3.8K | 8 | 6 | Regression |
| News | 31.6K | 8K | 45 | 14 | Regression |

- `Shuttle` (shu) is a multi-class classification dataset for extracting conditions in which automatic landing is preferred over manual control of the spacecraft.

- `Beijing` (Liang et al., 2015) is a regression dataset about PM2.5 air quality in the city of Beijing.

- `News` (Fernandes, 2015) is a regression dataset about online news articles to predict the number of shares in social networks.

The statistical information of datasets used in our experiments is in Table 25. #train, #test, #continuous, #categorical, and #class mean the number of training data, testing data, continuous columns, categorical columns, and class, respectively.

The raw data of 15 datasets are available online:

- `Credit`: https://www.kaggle.com/mlg-ulb/creditcardfraud (DbCL 1.0)

- `Default`: https://archive.ics.uci.edu/ml/datasets/default+of+credit+card+clients (CC BY 4.0)

- `HTRU`: https://archive.ics.uci.edu/ml/datasets/HTRU2 (CC BY 4.0)

- `Magic`: https://archive.ics.uci.edu/ml/datasets/magic+gamma+telescope (CC BY 4.0)

- `Phishing`: https://archive.ics.uci.edu/ml/datasets/phishing+websites (CC BY 4.0)

- `Shoppers`: https://archive.ics.uci.edu/ml/datasets/Online+Shoppers+Purchasing+Intention+Dataset (CC BY 4.0)

- `Spambase`: https://archive.ics.uci.edu/ml/datasets/spambase (CC BY 4.0)

- `Bean`: https://archive.ics.uci.edu/ml/datasets/Dry+Bean+Dataset (CC BY 4.0)

- `Contraceptive`: https://archive.ics.uci.edu/ml/datasets/Contraceptive+Method+Choice (CC BY 4.0)

- `Crowdsource`: https://archive.ics.uci.edu/ml/datasets/Crowdsourced+Mapping# (CC BY 4.0)

- `Obesity`: https://archive.ics.uci.edu/ml/datasets/Estimation+of+obesity+levels+based+on+eating+habits+and+physical+condition+ (CC BY 4.0)

- `Robot:` `https://archive.ics.uci.edu/ml/datasets/Wall-Following+Robot+Navigation+Data` (CC BY 4.0)

- `Shuttle:` `https://archive.ics.uci.edu/ml/datasets/Statlog+(Shuttle)` (CC BY 4.0)

- `Beijing:` `https://archive.ics.uci.edu/ml/datasets/Beijing+PM2.5+Data` (CC BY 4.0)

- `News:` `https://archive.ics.uci.edu/ml/datasets/online+news+popularity` (CC BY 4.0)

### H.3 EVALUATION METHODS

The reported scores for TSTR results in the paper are calculated as follows:

1. We download a dataset. If used previously, we use their train-test split. If not used before, we perform a new train-test split. The train-test split ratio is 80% and 20%, respectively.

2. Generate fake records which has the same number of records as the original training set for other fake data generation methods.

3. Using the training records from Step2, we train base classifiers/regressors to predict. We search the best hyperparameter set for each classifier/regressor. For validation, we use the original training data's task scores, e.g., F1 or $R^2$. In Table 26, considered hyperparameters and their candidate settings are summarized. We use DecisionTree, AdaBoost, Logistic Regression, MLP classifiers, RamdomForest, and XGBoost for binary classification tasks; DecisionTree, MLP classifiers, RandomForest, and XGBoost for multi-class classification tasks; MLP regressor, RandomForest, XGBoost, and Linear Regression for regression tasks.

4. Test the best performing classifiers/regressors with testing data. We use various evaluation metrics for rigorous evaluations as reported earlier.

We repeat the 4th step five times for all datasets. We then calculate the average score for each method and for each evaluation metric. Detailed metrics for our experiment are as follows:

1. Binary F1 for binary classification datasets: f1_score from sklearn.metrics after setting the 'average' option to 'binary'.

2. Macro F1 for multi-class classification datasets: f1_score from sklearn.metrics after setting the 'average' option to 'macro'.

3. Weighted-F1 for classification datasets: Weighted-F1 = $\sum_{i=0}^{N} w_i s_i$, where $N$ is the number of classes, the weight of $i$-th class $w_i$ is $\frac{1-p_i}{N-1}$, $p_i$ is the proportion of $i$-th class's cardinality in a total dataset, and score $s_i$ is a per-class F1 of $i$-th class (in a One-vs-Rest manner). This formula allows us to evaluate synthesized tables with more focus on mode collapse by giving a higher weight to a smaller class, which is more likely to be forgotten by the model.

4. AUROC: roc_auc_score from sklearn.metrics.

5. Coverage: compute_prdc from `https://github.com/clovaai/generative-evaluation-prdc`.

## I HYPERPARAMETERS

Hyperparameter settings for the best models are in Table 27. We have three SDE types, which are VE, VP, and sub-VP, and three layer types as shown in Appendix C: Concat, Squash, and Concatsquash. We use a learning rate in $\{2e-03, 2e-04\}$. We search for $\alpha_0$ and $\beta_0$, in total, with 9 combinations using $\alpha_0 = \{0.20, 0.25, 0.30\}$ and $\beta_0 = \{0.80, 0.90, 0.95\}$.

We also consider the hyperparameters for the fine-tuning process. To compute the exact log-probability with the Hutchinson's estimation (Hutchinson, 1989; Grathwohl et al., 2018), we use a Gaussian or a Rademacher distribution for $p(\epsilon)$, where $\epsilon$ is a noise vector. The fine-tuning epoch is $\{1, \ldots, 20\}$, and the fine-tuning learning rate is $\{2 \times 10^{-i} | i = \{4, 5, 6, 7\}\}$.

Table 26: Hyperparameters of the base classifiers/regressors

| Models | Hyperparameters | Values |
|---|---|---|
| DecisionTree | max_depth
min_samples_split
min_samples_leaf | 4, 8, 16, 32
2, 4, 8
1, 2, 4, 8 |
| AdaBoost | n_esimators | 10, 50, 100, 200 |
| Logistic Regression | solver
n_jobs
max_iter
C
tol | lbfgs
-1
10, 50, 100, 200
0.01, 0.1, 1.0
0.0001, 0.01, 0.1 |
| MLP | hidden_layer_sizes
max_iter
alpha | (100, ), (200, ), (100, 100)
50, 100
0.0001, 0.001 |
| RandomForest | max_depth
min_samples_split
min_samples_leaf
n_jobs | 8, 16, Inf
2, 4, 8
1, 2, 4, 8
-1 |
| XGBoost | n_estimators
min_child_weight
max_depth
gamma
nthread | 10, 50, 100
1, 10
5, 10, 20
0.0, 1.0
-1 |
| Linear Regression | - | - |

Table 27: The best hyperparameters used in Table 2

| Datasets | Hyperparameters for SPL of STaSy | | | | | | Hyperparameters for fine-tuning | |
|---|---|---|---|---|---|---|---|---|
| | SDE Type | Layer Type | Activation | Learn. Rate | $\alpha_0$ | $\beta_0$ | Hutchinson Type | Learn. Rate |
| Credit | VP | concatsquash | LeakyReLU | 2e-03 | 0.25 | 0.90 | Rademacher | 2e-05 |
| Default | VP | concatsquash | ReLU | 2e-03 | 0.30 | 0.90 | Rademacher | 2e-04 |
| HTRU | VE | concatsquash | LeakyReLU | 2e-04 | 0.25 | 0.95 | Rademacher | 2e-05 |
| Magic | sub-VP | squash | ReLU | 2e-03 | 0.3 | 0.95 | Rademacher | 2e-07 |
| Phishing | VP | squash | LeakyReLU | 2e-03 | 0.2 | 0.95 | Rademacher | 2e-07 |
| Shoppers | VE | concatsquash | ELU | 2e-03 | 0.30 | 0.95 | Rademacher | 2e-07 |
| Spambase | sub-VP | concat | LeakyReLU | 2e-04 | 0.2 | 0.95 | Rademacher | 2e-07 |
| Bean | VP | squash | ReLU | 2e-03 | 0.25 | 0.80 | Rademacher | 2e-05 |
| Contraceptive | VP | concatsquash | LeakyReLU | 2e-03 | 0.2 | 0.95 | Rademacher | 2e-07 |
| Crowdsource | VE | squash | LeakyReLU | 2e-03 | 0.25 | 0.90 | Gaussian | 2e-07 |
| Obesity | VP | squash | ELU | 2e-03 | 0.25 | 0.90 | Gaussian | 2e-05 |
| Robot | sub-VP | concat | ELU | 2e-03 | 0.3 | 0.95 | Rademacher | 2e-07 |
| Shuttle | sub-VP | squash | ReLU | 2e-03 | 0.3 | 0.95 | Rademacher | 2e-07 |
| Beijing | VE | concatsquash | LeakyReLU | 2e-03 | 0.25 | 0.90 | Rademacher | 2e-07 |
| News | VE | concatsquash | LeakyReLU | 2e-03 | 0.25 | 0.90 | Gaussian | 2e-07 |

## J ADDITIONAL VISUALIZATIONS

We show several visualizations that are missing in the main paper. In each subsection, we show column-wise histograms and t-SNE visualizations on `HTRU`, `Robot`, and `News`, respectively.

### J.1 ADDITIONAL VISUALIZATIONS IN `HTRU`

As shown in Figure 5, the fake data distributions of `TVAE`, `TableGAN`, and `RNODE` are dissimilar to the real data distributions, and these baselines fail to sample high-quality fake records. In Figure 6, `CTGAN`, `TableGAN`, `OCT-GAN`, and `RNODE` suffer from mode collapses as highlighted in red. In `STaSy`, however, the mode collapse problem is clearly alleviated, which means that `STaSy` is effective in enhancing the diversity.

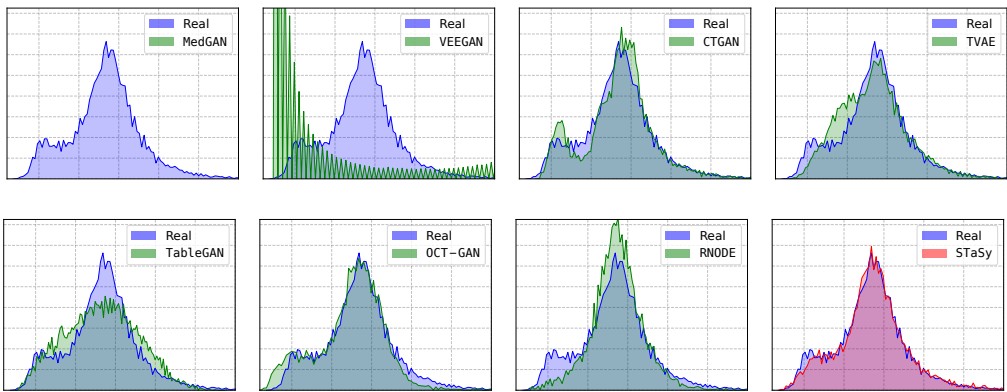

Figure 5: Histograms of values in the *excess kurtosis* column of `HTRU`

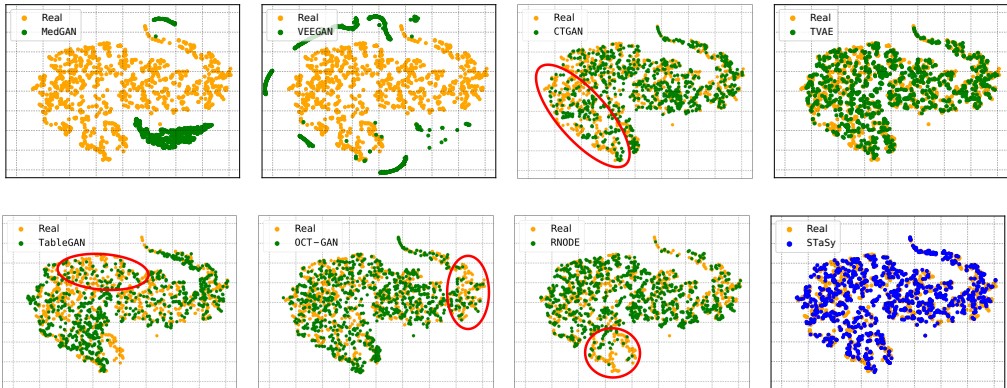

Figure 6: t-SNE visualizations of fake and original records in `HTRU`

## J.2 ADDITIONAL VISUALIZATIONS IN ROBOT

In `Robot`, `TVAE`, `OCT-GAN`, and `STaSy` generate relatively similar distribution to that of real data as shown in Figure 7. In Figure 8, `CTGAN`, `TableGAN`, and `RNODE` generate out-of-distribution records as highlighted in red, and `TVAE` and `OCT-GAN` suffer from mode collapses. Our proposed methods generate the most diverse records among various methods.

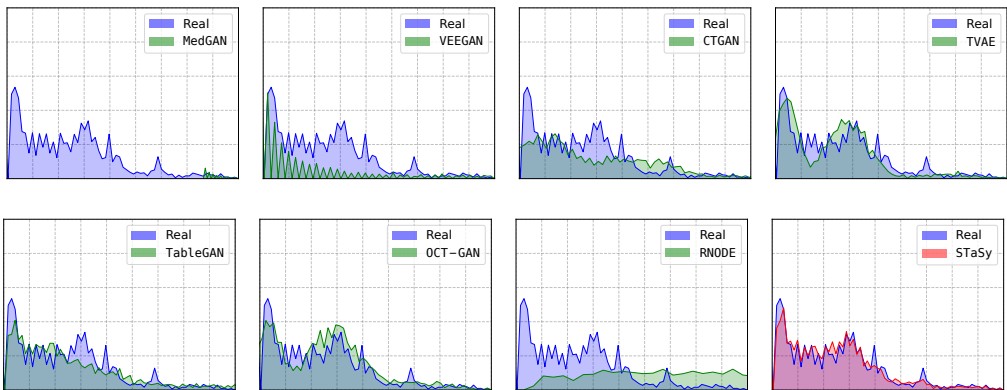

Figure 7: Histograms of values in the *ultrasound sensor* column of `Robot`

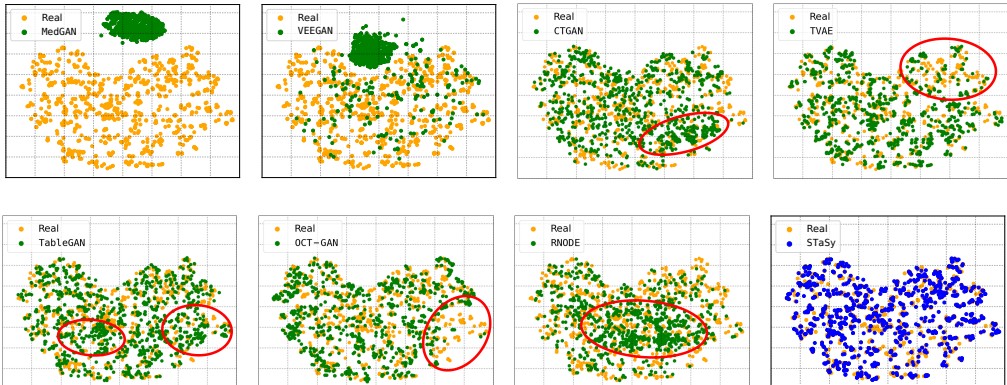

Figure 8: t-SNE visualizations of fake and original records in `Robot`

## J.3 ADDITIONAL VISUALIZATIONS IN NEWS

In News, the fake data by STaSy shows more reliable column-wise histogram than others in Figure 9. As shown in Figure 10, all methods except for TableGAN and RNODE well generate fake records.

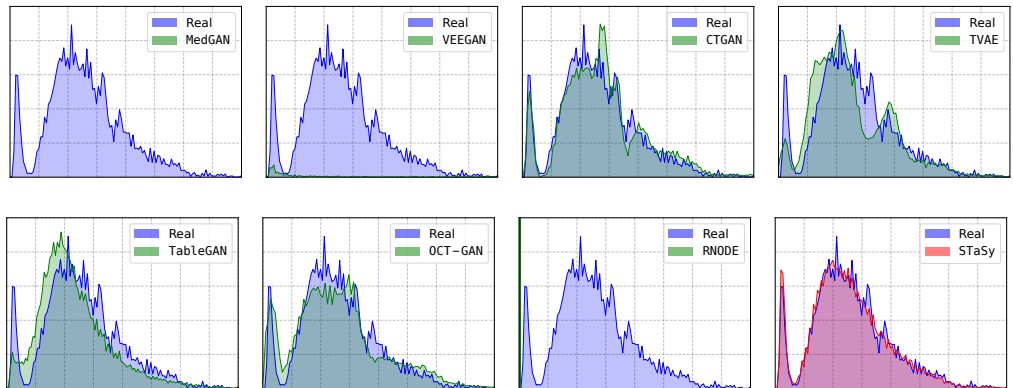

Figure 9: Histograms of values in the *min of best keyword* column of News

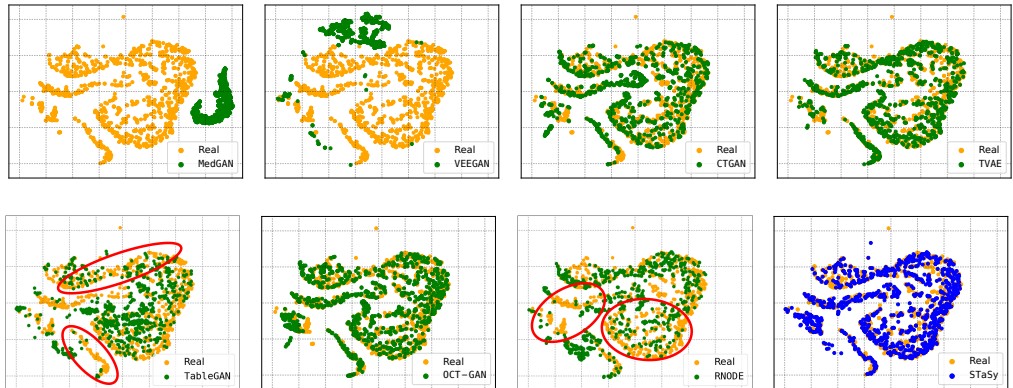

Figure 10: t-SNE visualizations of fake and original records in News

