# OpenReview forum: "STaSy: Score-based Tabular data Synthesis"
_ICLR.cc/2023/Conference — ICLR 2023 notable top 25%_

### Official Review · Reviewer_xhm1 · 2022-10-21

**Confidence:** 3
**Correctness:** 2
**Technical Novelty And Significance:** 3
**Empirical Novelty And Significance:** 3
**Recommendation:** 5

**Clarity, Quality, Novelty And Reproducibility:**

The paper is mostly clearly written, except for the above comments. It represents a novel approach. The authors used public datasets and provided the code, so the results should be reproducible.

**Strength And Weaknesses:**

The paper shows an interesting new approach to an important problem, and it is clearly written. However, I have several concerns:
- The first comparison of quality, diversity and run-time was only done with 5 out of the 7 previous methods. Why weren't all of them checked?
- It is unclear which data was used for the hyper-parameter tuning on each dataset (no validation set was defined and no cross-validation was mentioned).
- The hyper-parameter search for the classification/regression models is very limited. What are the state-of-the-art results obtained for these datasets? Are they close to the presented results?
- It would be interesting to see how the results of one state-of-the-art model, such as XGBoost, are impacted by using different types of artificial data (or real data), while trying either just the default hyperparameters, or a large set of possible hyperparameters.
- In the Appendix, the results on the artificial data are sometimes better than the results on the real data - what could explain this?
- Why the comparison with SOS was only done using three datasets? (in the Appendix)



**Summary Of The Paper:**

This paper considers the problem of generating tabular data with a similar distribution to an existing dataset. It provides a technique for generating such artificial data, using SGM (score-based generative models). The results are further improved by using self-paced learning and fine-tuning of the model per dataset. The new method is compared with 5 previously proposed methods, showing that the new method generates data that is harder to distinguish from the real data and covers a larger portion of the distribution (about 1.5x better), but runs significantly slower than most previous methods (two were slower, three were ~100x faster). The paper compares modeling results on the artificial data with 7 previous methods, as well as with real data (the test data is the same real data in all cases). The evaluation is done on 15 datasets, including binary classification, multiclass classification and regression. The training is done using several models (e.g., XGBoost, MLP, Decision tree, Adaboost, Logistic Regreesion, Linear Regression), a hyperparameter search is done for each of them and the best result is used (average of 5 runs). The paper shows that training the models on the new artificial data yields better predictions than on the data generated using previous methods. The results are close to the results obtained when training on real data. The paper additionally presents illustrations that show examples where previous methods generated out of distribution samples, partial coverage or differences in distribution, while the new method performed well.


**Summary Of The Review:**

This paper presents an interesting technique for an important problem. It could be useful, for example, when the actual data is private and cannot be shared. However, I have several concerns about the evaluation and about presentation, as described in the weaknesses paragraph above. Based on the authors response I will reconsider my score.

---

### Official Review · Reviewer_A3vU · 2022-10-24

**Confidence:** 2
**Correctness:** 4
**Technical Novelty And Significance:** 4
**Empirical Novelty And Significance:** 4
**Recommendation:** 8

**Clarity, Quality, Novelty And Reproducibility:**

Clarity
-------
Paper is very well written.  Some minor comments:
- In Figure 3, please list what feature distributions are being plotted

-Displaying the smoothed histograms of losses in Figure 1 is potentially misleading without the time element.

-"Our proposed score network architecture is in Appendix C." <- Please briefly describe the architecture in the main text

-"We set α0 and β0 in such a way that more than 80% of the training records are included in the learning process from the beginning." <- Please state exactly what this way is (if too long, it can always go in the appendix).

-Please report runtime in Table 2.

-In Algorithm 1, how are \theta and \mathbf{v} initiliazed?

-"Since STaSy can be converted to an oversampling method following the design guidance of SOS," <- Could you describe this design guidance?

Quality
--------
High-quality paper; the proposed algorithm is intuitive and the included experiments clearly demonstrate superior performance to existing tabular synthesizer.  The ablation studies were also very effective at showing the contributions of SPE and fine-tuning components.

Novelty
---------
The paper contains several novel contributions: application of diffusion models for general tabular data synthesis (as well as demonstration of the effectiveness of this as a staight-forward approach), a novel SPE algorithm, and the full STaSy model

Reproducibility
------------------
Author's have included their source code to reproduce all featured experiments.


**Strength And Weaknesses:**

Strengths
------------
-The paper's contributions are significant: 1) Demonstration of score-based generative modeling as a superior means of tabular synthesis (compared to direct GAN approaches), (2) A new novel SPE algorithm, (3) A new tabular synthesizer, STaSy, which demonstrates impressive performance on the evaluated tasks

-Extensive results and ablation studies are compelling and convincing to show the utility of the SPE and fine-tuning algorithms
-The paper is well written

-The use of simple feature pre/post-processors is a big win, as mixture-model based pre-processing (i.e., those used in CTGAN/TVAE) are extremely expensive for large-scale data

Weaknesses
---------------
-The runtime of STaSy is a major concern.  E.g., in Table 1, STaSy runtime is more than an order of magnitude slower than CTGAN, which itself struggles on high-dimensional data (e.g., it is not scalable for feature dimensions >150).  Could the authors comment on how STaSy would fare in such high-dimensional regimes?  Since runtime is such an important practical metric, it would greatly strengthen the paper if the authors could further discuss how STaSy's runtime could be decreased.  E.g., could you speak to the possibility of using [1] to avoid Gaussian sampling in the reverse process and speed up sampling time?

-While well-written, there is a lot of content in the main text, as well as the appendix.

-"we follow the "train on synthetic, test on real (TSTR)" framework": this has become a widely adapted framework in tabular synthesizer papers.  However, it is impractical, and does not actually show how such synthesizers may be realistically leveraged in practice.  E.g., while privacy arguments are commonly made, the most important categorical features cannot be synthesized directly (e.g., a new email would not be generated by STaSy, CTGAN, VEEGAN, etc.).  Yet, it is impressive that STaSy is the first method which achieves identical testing performance given purely synthetic training data (as well as being a rare example where performance under purely real training data is even reported).  A much better demonstration of utility is data augmentation.  I think including more discussion of the oversampling result in the appendix would further show how impressive this method is.

[1] Xiao, Zhisheng, Karsten Kreis, and Arash Vahdat. "Tackling the
generative learning trilemma with denoising diffusion gans." arXiv
preprint arXiv:2112.07804 (2021).

**Summary Of The Paper:**

The authors propose the use of score-based generative modeling (i.e., diffusion models) for general tabular synthesis, and demonstrate the effectiveness this approach across a large number of tabular datasets and synthesizers (e.g., CTGAN, VEEGAN, TableGAN).  Building on this approach, the authors introduce a novel self-paced learning algorithm following by fine-tuning (the combination of which is called STaSy), which further improves upon the direct application of score-based generative modeling.  Extensive studies and results are shown, demonstrating the the strengths of the additional components of STaSy, which achieves SOTA in all performance evaluations (but is an order of magnitude slower than other GAN-based approaches during test time).

**Summary Of The Review:**

High quality paper with minor comments and suggestions for improvement.  Only major concern is with the runtime of the method, suggest authors address this in the main paper.  I recommend acceptance.

---

### Official Review · Reviewer_AiyZ · 2022-10-26

**Confidence:** 3
**Correctness:** 3
**Technical Novelty And Significance:** 3
**Empirical Novelty And Significance:** 3
**Recommendation:** 8

**Clarity, Quality, Novelty And Reproducibility:**

Although most of the paper is clearly written, the following parts are unclear.
1. The problem of naive stasy is not explained well. What I have gleaned from Fig 1 is that the loss values of naive stasy are not concentrated around zero, which implies underfitting occurs. Why does this happen? Is it because of the loss function or learning dynamics?
1. Related to the above question, the motivation for introducing self-paced learning is not explained. I know self-paced learning is introduced to alleviate the above problem, but it is not sure why self-paced learning can mitigate the problem.

The quality is high. They evaluated the proposed model on many datasets with several baselines.

As far as I know, this is the first SGM for tabular data, and it is novel.

In terms of reproducibility, they provide the source code.

**Strength And Weaknesses:**

Strengths
- A reasonable adaptation of SGM for tabular data.
- The performance is quite nice.
- The performance is evaluated in both quantitative and qualitative ways.
- The paper provides rich information (comparison with another SGM model, details of experiments, etc) in Appendix.

Weaknesses
- The motivation for introducing self-paced learning is not clearly explained.
- There are additional hyperparameters to tune in the proposed method.

**Summary Of The Paper:**

This paper studies a score-based generative model (SGM) for tabular data. They found there's a difficulty in adopting SGM to tabular modeling that the training process is unstable. Hence they introduced a self-paced learning algorithm, a variation of curriculum learning, to train SGM. In experiments, the proposed method outperforms other generative models, including GAN-based models, with several real datasets.

**Summary Of The Review:**

A solid SGM is proposed for tabular data synthesis. Although there's room to improve the description of the method/algorithm derivation, the paper's quality is sufficient for publication.

---

### Official Review · Reviewer_dEhg · 2022-11-05

**Confidence:** 4
**Correctness:** 4
**Technical Novelty And Significance:** 3
**Empirical Novelty And Significance:** 4
**Recommendation:** 8

**Clarity, Quality, Novelty And Reproducibility:**

Paper is very clearly written. Experiments are comprehensive. Code is provided so reproducibility should not be a concern. Novelty is not perfect since this paper combines several existing techniques known in other data modalities.

**Strength And Weaknesses:**

## Strength

1. Tabular data is a very important data modality. Improving the performance on tabular data generation is very valuable for privacy protection, data augmentation, etc.

2. Paper is clearly written. The proposed method is straightforward, properly motivated, and easy to implement.

3. Experiments demonstrate the efficacy of StaSy at generating tabular data from multiple directions. Ablation study clearly shows the advantages of self-paced learning and fine-tuning.

## Weaknesses

1. The finetuning process as given in Algorithm 1 uses log likelihood to compute the threshold, but it seems that authors actually used equation (6) instead of log-likelihoods. Why not change Algorithm 1 to use equation (6) instead of log-likelihood? You can mention in text that log-likelihood is also a natural choice but turns out to perform worse in practice.

2. Some minor writing issues need to be fixed:

  * The bottom line on page 1: "However" is an awkward word to be there.

  * Right below equation (3): remove "the" before "denoising score matching"

  * First line in Theorem 1: replace "is" with "be.

  * Also in Theorem 1: the optimal solution is not defined, but proved in equation (9)

  * Section 3.4: "However" is again a bad choice of word.

**Summary Of The Paper:**

This work proposes to generate tabular datasets with score-based generative models. For improved performance, authors apply self-paced learning to create a curriculum where harder data points are trained later than easier ones. A final fine-tuning process is proposed to further improve the performance on hard data points. Experiments demonstrate that the proposed Stasy method outperforms existing generative models in creating realistic tabular data samples that can fool classifiers. Ablation study corroborates the efficacy of self-paced learning and fine-tuning.

**Summary Of The Review:**

Nice written paper with simple but effective approaches and strong experimental results.

---

### Author Response · Authors · 2023-05-29
**Code is available.**

The code for our experiments is available at https://github.com/JayoungKim408/STaSy.

In the repository, we also provide a checkpoint for $\texttt{Shoppers}$.

If you have any questions or would like to discuss anything with us, please feel free to contact us via any available means, such as through GitHub or the email provided in the manuscript.

---

### Decision · Program_Chairs · 2023-01-20

**Decision:**

Accept: notable-top-25%

**Justification For Why Not Higher Score:**

I'm not sure how interesting this would be to a general ICLR audience, but if the senior ACs believe it should be an oral presentation, I would not object.

**Justification For Why Not Lower Score:**

The authors propose a diffusion model for a new modality, incorporate novel techniques to improve tabular data generation, and achieve strong results. Moreover, all the reviewers agree that the paper should be accepted.

**Metareview: Summary, Strengths And Weaknesses:**

The authors propose a score-based method for tabular data synthesis. They show that a naive approach generate quite poor results, and show how, with changes to the architecture, training with self-paced learning, and fine-tuning can significantly improve the model. The experimental results are strong, the ablations well motivated.

**Note From Pc:**

if the above contains the word "oral" or "spotlight" please see: "oral" presentation means -> notable-top-5% and "spotlight" means -> notable-top-25%. As stated in our emails, we are disassociating presentation type from AC recommendations